

# Western Mediterranean hydro-climatic consequences of Holocene iceberg advances (Bond events)

Christoph Zielhofer[1], Anne Köhler[1], Steffen Mischke[2], Abdelfattah Benkaddour[3], Abdeslam Mikdad[4], William J. Fletcher[5]

[1]Institute of Geography, Chair of Physical Geography, Leipzig University, Leipzig, 04103, Germany
[2]Faculty of Earth Sciences, University of Iceland, Reykjavík, 101, Iceland
[3]Department of Earth Sciences, Faculty of Science and Technology, Cadi Ayyad University, Marrakech-Guéliz, Morocco
[4]Institut National des Sciences de l'Archéologie et du Patrimoine, Rabat, Morocco
[5]Department of Geography, School of Environment, Education and Development, University of Manchester, Manchester, M13 9PL, United Kingdom

*Correspondence to*: Christoph Zielhofer (zielhofer@uni-leipzig.de)

**Abstract.** Gerald C. Bond established a Holocene series of North Atlantic ice rafted debris events based on quartz and hematite stained grains recovered from subpolar North Atlantic marine cores. These so-called 'Bond events' document nine large-scale and multi-centennial North-Atlantic cooling phases that might be linked to a reduced thermohaline circulation. Regardless of the high prominence of the Holocene North Atlantic ice rafted debris record, there are critical scientific comments on the study: the Holocene Bond curve has not yet been replicated in other marine archives of the North Atlantic and there exist only very few palaeo-climatic studies that indicate all individual Bond events in their own record. Therefore, evidence for consistent hydro-climatic teleconnections between the subpolar North Atlantic and distant regions is not clear. In this context, the Western Mediterranean region reveals key hydro-climatic sites for the reconstruction of a teleconnection with the subpolar North Atlantic. In particular, variability of Western Mediterranean winter precipitation might be the result of atmosphere-ocean coupled processes in the outer-tropical North Atlantic realm.

Based on an improved Holocene $\delta^{18}O$ record from Lake Sidi Ali (Middle Atlas, Morocco) we correlate Western Mediterranean precipitation anomalies with North Atlantic Bond events to identify a probable teleconnection between Western Mediterranean winter rains and subpolar North Atlantic cooling phases. Our data show a noticeable positive correlation between Western Mediterranean winter rain minima and Bond events during the Early Holocene and an opposite pattern during the Late Holocene. There is evidence for an enduring hydro-climatic change in the overall Atlantic atmosphere-ocean system and the response to external forcing during the Mid-Holocene. Regarding a potential climatic anomaly around 4.2 ka (Bond event 3) in the Western Mediterranean, a centennial-scale winter rain maximum is generally in phase with the overall pattern of alternating 'wet and cool' and 'dry and warm' intervals during the last 5,000 years.





# 1 Introduction

Gerald C. Bond reconstructed a Holocene series of North Atlantic ice-rafting events (Bond et al. 1997, 2001) based on the numbers of counted quartz and haematite stained grains in marine cores recovered from the subpolar North Atlantic (Fig. 1a). These so-called 'Bond events' document nine (Fig. 2c) large-scale and multi-centennial North-Atlantic cooling phases that

5    might be linked to a reduced thermohaline circulation in the North Atlantic. Due to attested large-scale atmosphere-ocean-linked teleconnections, an increasing number of palaeoclimatologists relate Bond events with chronologically in-phase climatic anomalies all over the world. To this day, the manuscript about North Atlantic ice rafted debris events (Bond et al. 2001) is one of the most cited papers on the Holocene climate history (2630 citations, Google Scholar, 2018).

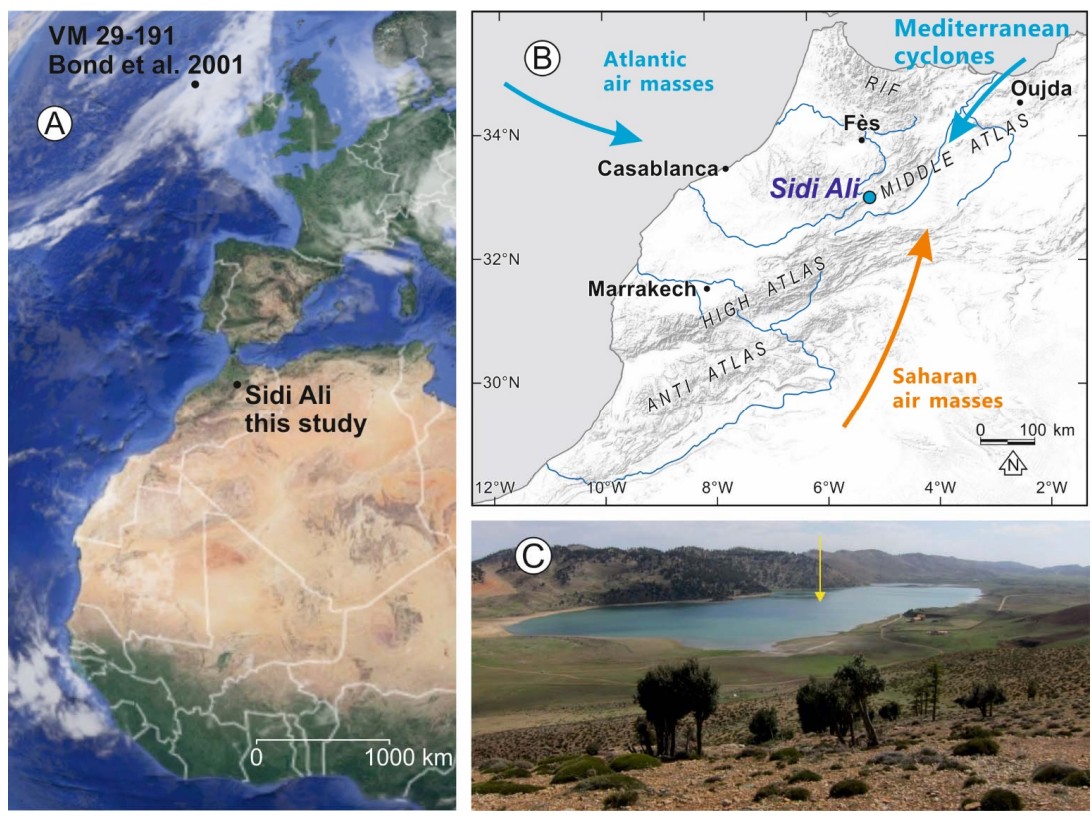

**Figure 1: Geographical setting.** A) Positions of the stacked ice rafted debris record (MC52 + VM29-191, Bond et al. 2001) and the improved Sidi Ali oxygen stable isotope record (this study); B) Lake Sidi Ali in the Moroccan Middle Atlas, arrows indicate impacts of different air masses on Holocene climate history; C) Lake Sidi Ali, the yellow arrow shows the core position.



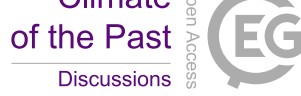

Regardless of the high prominence of the Holocene North Atlantic ice rafted debris record, there are also numerous critical scientific comments on the study: **a)** As far as the authors are aware, the stacked Holocene Bond curve has not yet been replicated in other marine archives of the North Atlantic, or could not be reconstructed. **b)** Some marine geologists question whether sand grains in these marine cores represent lithological material transported by rafted icebergs (Sejrup et al., 2011).

**c)** Bond et al. (2001) postulated a quasiperiodic, "1500-year" cycle in the Holocene ice rafted debris record. However, a Holocene 1,500 cycle remains controversial in the scientific community (Darby et al., 2012; Obrochta et al., 2012). **d)** The forcing mechanisms for the multi-centennial to millennial-scale ice rafted debris events are not clear (Bügelmayer-Blaschek et al. 2016). **e)** Furthermore, there exist only very few palaeoclimatic studies (Cheng et al., 2015; Smith et al., 2016) that indicate all individual Bond events in their own record. Therefore, evidence for consistent hydro-climatic teleconnections

between the subpolar North Atlantic and distant regions is not clear for the entire Holocene.

In this context, the Western Mediterranean region features key hydro-climatic sites for the reconstruction of a potential teleconnection with the subpolar North Atlantic. In particular, Western Mediterranean variability of Holocene winter precipitation is the result of large-scale atmosphere-ocean coupled processes in the outer-tropical North Atlantic realm (Lamb

et al., 1995; Trouet et al., 2009; Wassenburg et al., 2013, 2016; Zielhofer et al. 2017a). In addition to indications from continuous hydro-climatic archives, Western Mediterranean alluvial archives show a link with Bond events. Fluvial geomorphologists identify Holocene flood intervals that chronologically match with peaks in the ice-rafted debris record. This is the case for Western Mediterranean fluvial records in Morocco and Tunisia (Faust et al., 2004; Zielhofer and Faust, 2008; Zielhofer et al., 2010) but also for fluvial records in Western Mediterranean Europe (Benito et al. 2008; Wolf et al., 2013;

Benito et al. 2015a, 2015b). Furthermore, Western Mediterranean proxy data for prehistoric human occupation, such as [14]C cumulative probability plots from archaeological databases, show a probable linkage with ice rafted debris events (Zielhofer et al., 2008; Linstädter, 2016).

Although many Western Mediterranean hydro-climatic records attest coincidences with Bond events, the forcing mechanisms

and chronological correlations are not clear: **a)** In the Western Mediterranean multiple studies show that Holocene humidity changes are locally variable and with reverse directions (Morellón et al., 2018). So, whereas palaeoecological studies from the Pyrenees indicate environmental conditions that are more humid during Mid- and Late Holocene North Atlantic cooling events (Pélachs et al., 2011), pollen records from the Western Mediterranean Sea (Fletcher et al., 2013) and a prominent $\delta^{18}$O speleothem record from Northern Spain (Smith et al., 2016) show arid intervals. **b)** Standard age errors of [14]C and OSL dating

techniques but also the non-continuous and non-linear deposition pattern of many terrestrial archives, like flood deposits respectively (Faust and Wolf, 2017), do not enable accurate age models and a direct synchronisation with Bond events.

In a previous manuscript we presented a stable oxygen isotope record of Holocene benthic ostracods from Lake Sidi Ali in the Middle Atlas, Morocco that indicates multi-centennial to millennial intervals of Western Mediterranean winter rain minima



during the last 12,000 years (Zielhofer et al., 2017a). Here, $\delta^{18}O$ maxima correspond with winter rain minima. However, the mean chronological resolution of the previous stable oxygen isotope record is ~130 years and only allows a limited comparison with palaeoecological proxy data from the same core. In the present manuscript, the chronological resolution of the Sidi Ali $\delta^{18}O$ record is improved. We aim to compare the higher resolution $\delta^{18}O$ data with published *Cedrus* pollen and microcharcoal

records from the same core (Campbell et al., 2017). The direct comparison of palaeoclimatic and palaeoecological data from the same core enables a multi-proxy interpretation with less age uncertainties that allows a profound understanding of the Western Mediterranean hydro-climate history. Furthermore, we correlate for the first time the newly established Sidi Ali $\delta^{18}O$ record with the North Atlantic ice rafting debris record (Bond et al., 2001) to identify a probable teleconnection between Western Mediterranean winter rains and ocean-atmosphere coupled cooling phases in the subpolar North Atlantic. Finally, we

provide an analysis of the Western Mediterranean hydro-climate during the 4.2 ka Climatic Event (Bond event) that is in the focus of the present "Climate of the Past" special issue.

## 2 Study Area

### 2.1 Lake Sidi Ali geographical and hydro-climatic setting

The geographical position of the karstic Lake Sidi Ali in the Middle Atlas (33° 03' N, 5° 00' W, 2080 m a.s.l.) is within the mountainous desert margin of Morocco between the subhumid Mediterranean climate in the North and the arid Saharan climate in the South (Fig. 1b). The mean annual precipitation at Lake Sidi Ali is about 430 mm with a mean annual temperature of 10.3°C (mean JJA maximum, 32.5°C; mean DJF minimum, -8.4°C) and a dry season lasting from June to September (Zielhofer et al. 2017b). The current hydro-climate at Sidi Ali is characterised by Atlantic cyclones during the winter season with a strong

impact of the present-day North Atlantic Oscillation (NAO) providing more precipitation during NAO negative stages (Hurrell, 1995; Hurrell et al., 2003). In contrast, Mediterranean cyclones are more associated with rainfall during spring and autumn (Knippertz et al., 2003). The surrounding forest vegetation, consisting of evergreen oak (*Quercus rotundifolia*) and Atlantic cedar (*Cedrus atlantica*), is strongly degraded due to overgrazing. The lake lies within a closed basin of approx. 14 km² and has a varying surface area between 2.0 and 2.8 km² (Sayad et al., 2011). During late summer 2012, Lake Sidi Ali waters had

$\delta^{18}O$ values between +1.21 and +2.57‰ vs. SMOW, a surface temperature of 18.5°C and a lake bottom temperature of 8.7°C. The surface $\delta^{18}O$ values are higher than those of bottom waters indicating the evaporative enrichment during summer stratification.

### 2.2 Lake Sidi Ali core recovery and chronology

At the deepest part of Lake Sidi Ali our research group conducted a drilling campaign in September 2012. A 19.56 m sequence from a single borehole was recovered using an UWITEC piston corer. The sediments consist of faintly laminated, organic silts with some aquatic macrofossils including ostracods. The sequence is continuous without any hiatus (Zielhofer et al. 2017a). Our Bayesian age model is based on 26 AMS ${}^{14}C$ dates on pollen concentrates and terrestrial plant remains and ${}^{210}Pb$ and ${}^{137}Cs$



radiometric dating (Fletcher et al., 2017). The age model reveals a coherent robust chronology, which provides a continuous record for the last 12,000 years (Fig. S1).

### 3 Methods: oxygen isotopes of ostracod shells

5 We add 82 new samples of adult ostracod shell material from the closely related species *Fabaeformiscandona* sp. and *Candona* sp. to improve the chronological resolution of the previous oxygen isotope record (Zielhofer et al., 2017a). For ostracod sampling, 8g dry sediment was freeze-dried and treated with 3 % $H_2O_2$. Afterwards, the samples were wet-sieved with a mesh size of 250 μm. The residues were dried at 50° C. Then, ostracod shells were picked under a binocular microscope. Four to six adult shells (about 20 μg) were used for oxygen isotope analyses. Shells were reacted with 105 % phosphoric acid at 70 °C

10 using a Kiel IV online carbonate preparation line connected to a MAT 253 mass spectrometer. Reproducibility was checked by replicate analysis of NBS19 and was better than ±0.06 ‰ (1 σ) for $\delta^{18}O$ values. The newly conducted oxygen isotope data (vs. PDB) were integrated in the existing Sidi Ali age model (Fletcher et al., 2017). Values of the ice rafted debris record (Bond et al., 2001) were interpolated to respective ages of Sidi Ali oxygen isotope data points. The running correlation between the ice rafted debris record and the improved Sidi Ali oxygen isotope data bases on 25 surrounding values.

### 4 Results and Discussion

The improved Sidi Ali $\delta^{18}O$ record (grey line in Fig. 2d) consists of 168 data points for the last 12,000 years and provides a mean chronological resolution of 71.4 years. The chronological frame encompasses the entire Holocene and the last 400 years of the Late Glacial. Due to the scattering of the original data, a three-point running mean was computed (black line in Fig. 2d)

20 for a better visualisation of the Holocene trajectory and the multi-centennial to millennial trends.



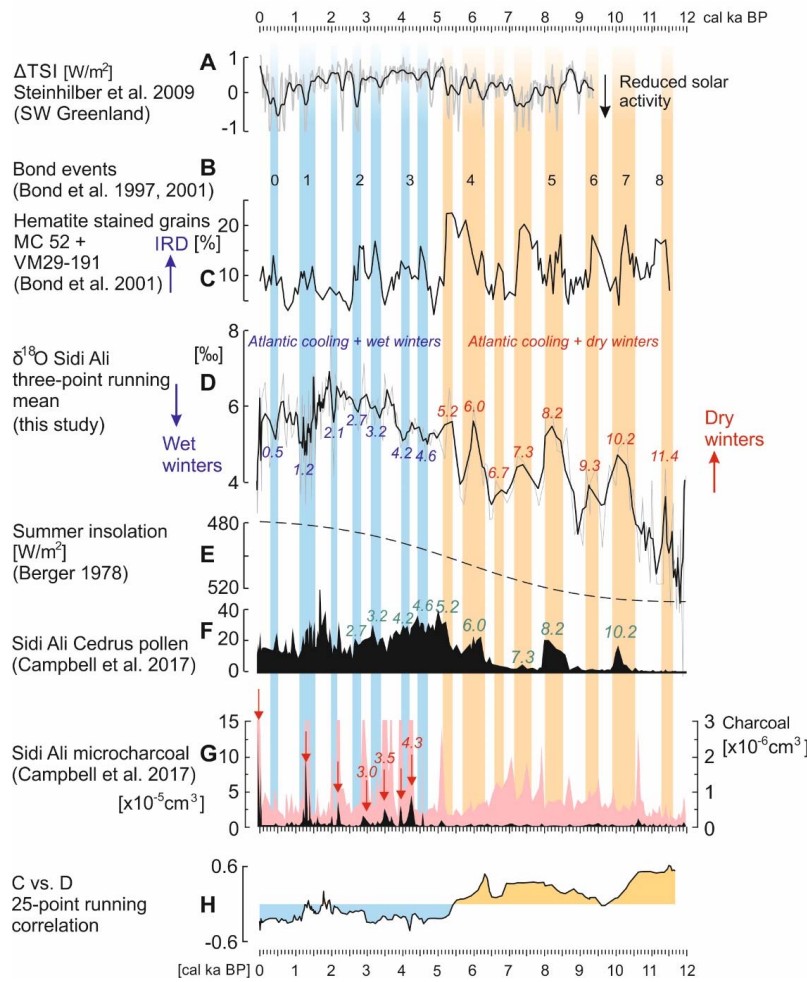

**Figure 2: Holocene North Atlantic ice-rafted debris record versus Western Mediterranean hydro-climatic record:** A) Total solar irradiance (ΔTSI, Steinhilber et al., 2009); B) Holocene Bond events 0 to 8 derived from Bond et al. (1997, 2001); C) Ice-rafted debris (IRD) record based on hematite stained grains of stacked MC52 and VM29-191 cores from the subpolar North Atlantic (Bond et al. 2001); D) Improved Sidi Ali δ18O record from closely related species *Fabaeformiscandona* sp. and *Candona* sp (this study). The grey line represents the original data, the black line shows the three-point running mean. Blue/red numbers and pale blue/orange bars indicate North Atlantic cooling events and wet/dry winters in the Western Mediterranean; E) Summer insolation (65°N, June, Berger, 1978) (note reversed axis); F) Sidi Ali *Cedrus* pollen record (Campbell et al., 2017). The green numbers represent noticeable maxima in *Cedrus* pollen; G) Sidi Ali fire proxies of microcharcoal concentration (CHAC) and microcharcoal accumulation rate (CHAR) (Campbell et al., 2017). The red arrows and red numbers indicate Late Holocene phases of increased fire activity; H) 25-point running correlation between the IRD record (C) and the Sidi Ali δ18O record (D).



### 4.1 Western Mediterranean hydro-climate anomalies during the Holocene

*Full range of the δ¹⁸O ostracod record from the Sidi Ali core*

In consideration of present $\delta^{18}O$ values and temperatures of Lake Side Ali waters (Zielhofer et al. 2017a) and according to the
equation of Friedman and O'Neil (1977), carbonate formed in the modern Lake Sidi Ali waters should have $\delta^{18}O$ values

between +0.5 and +4 ‰. In contrast, carbonates formed in nearby freshwater springs and streams currently reveal much lower
values between -10 and -4 ‰, indicating that the higher values in the lake waters are significantly affected by evaporation
(Benkaddour et al., 2005, Zielhofer et al. 2017a). The computed $\delta^{18}O$ values between +0.5 and +4 ‰ for carbonates in modern
lake waters fit well with the youngest $\delta^{18}O$ value from Sidi Ali ostracod shells that attain +3.8 ‰. The full ostracod record
ranges from -1.1 to +8.1 ‰ (Fig. 2c). These $\delta^{18}O$ values were always higher than the computed values of the current freshwater

springs, providing evidence for an always closed-basin lake during the recorded period.

*Long-term Holocene change of the Sidi Ali δ¹⁸O record*

In the Sidi Ali core, the $\delta^{18}O$ values increase from approx. +3 to +5 ‰ in the Early Holocene to values from +5 to +7 ‰ in the
Late Holocene (Fig. 2d). As the most straightforward scenario for a subhumid, closed basin (Roberts et al., 2008) this implies

a decrease of the precipitation/evaporation ratio with generally more arid conditions towards the Late Holocene. This
corresponds with Sidi Ali diatom, TOC and carbonate records that indicate high lake levels during the Early Holocene and
lower levels at later stages (Zielhofer et al., 2017a). Furthermore, low microcharcoal counts were observed in the Early
Holocene record of Lake Sidi Ali (Fig. 2g; Campbell et al., 2017) that support the scenario of an overall more humid climate
at that time.

These findings seem to contradict the *Cedrus* pollen record from Lake Sidi Ali (Fig. 2f; Campbell et al., 2017) that shows a
low or even missing occurrence of cedars during the Early Holocene, indicating reduced moisture availability at that time.
However, reduced moisture availability for the cool-preferring cedar seems to be the result of enhanced summer heat during
the Early Holocene. Due to their shallow roots, cedars are vulnerable to summer heat in contrast to the warm tolerant and deep-

rooting evergreen oaks that dominate the Sidi Ali pollen record during the Early Holocene (Campbell et al., 2017). In this
context, we attest summer temperature-driven drought stress and not winter precipitation as the limiting factor for the long-
term trend of Holocene cedar occurrence in the Middle Atlas. This inference is in good agreement with the orbital-forced
summer insolation maximum during the Early Holocene (Fig. 2e; Berger 1978) and the chironomid-based summer temperature
reconstructions that indicate enhanced Mediterranean summer temperatures during the Early Holocene as well (Samartin et

al., 2017). Furthermore, we worked out that enhanced $\delta^{18}O$ values are also the result of a specific origin and seasonality of the
precipitation-bearing air masses. Precipitation from springtime Mediterranean cyclones reveals higher $\delta^{18}O$ values than
Atlantic winter rains (Zielhofer et al. 2017a). In summary, Sidi Ali multi-proxy interpretation attest in the long-term orbital
trend a summer-warm climate rich in Atlantic winter rains during the Early Holocene in contrast to a summer-cool climate

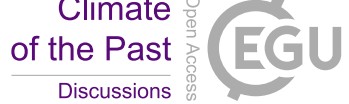



with reduced Atlantic winter rain and relatively increased occurrences of springtime Mediterranean cyclones in the Late Holocene.

*Millennial- to multi-centennial Holocene hydro-climatic variability in the Western Mediterranean*

The Sidi Ali $\delta^{18}O$ curve displays noticeable multi-centennial to millennial peaks for the last 12,000 years (Fig. 2d). During the Early and first half of the Mid-Holocene $\delta^{18}O$ maxima correspond with enhanced occurrences of *Cedrus* pollen (Fig. 2f; Campbell et al., 2017), this is especially the case at 10.2, 8.2, 7.3 and 6.0 cal ka BP. Whereas increased $\delta^{18}O$ maxima indicate reduced winter rain, synchronous increases in *Cedrus* pollen might be the result of enhanced moisture availability during summer. Hence, we infer the long-term climate change from a winter-wet and summer-warm climate towards drier winters

and cooler summers also for Early and Mid-Holocene climatic shifts at multi-centennial to millennial time-scales.

However, the relationship between the *Cedrus* and $\delta^{18}O$ values changes in the Mid-Holocene and passes to a contrasting but more typical pattern of an increased P/E ratio with $\delta^{18}O$ troughs and peaks in the *Cedrus* curve (pale blue bars in Figs. 2d, 2f). This is especially striking around 4.2, 3.2 and 2.7 cal ka BP and is supported by the pattern of the microcharcoal record at that

time. Peaks in microcharcoal at 4.3, 3.5 and 3.0 cal ka BP (red arrows in Fig. 2) parallel with $\delta^{18}O$ peaks indicating higher fire activity during phases that are more arid. However, during the last two millennia human impact might have strengthened fire activity due to the occurrence of distinct micocharcoal peaks at wetter phases.

## 4.2 Western Mediterranean winter rainfall anomalies parallel with Bond events

*Coupling with the North Atlantic ice rafted debris record*
Until here, we underpin our interpretation of the improved Sidi Ali $\delta^{18}O$ record only with proxy data from the same core to prevent potential misinterpretations due to age inaccuracies. On this basis, we are able to provide the Sidi Ali $\delta^{18}O$ record as a robust proxy for hydro-climatic changes in the Western Mediterranean, in particular for North Atlantic-forced changes in winter rain. Here, peaks in the $\delta^{18}O$ curve are interpreted as winter rain minima. In a next step, the Sidi Ali $\delta^{18}O$ record is

compared with the prominent subpolar North Atlantic ice rafted debris record from stacked MC52 and VM29-191 marine cores (Fig. 2c; Bond et al., 2001). Although we have to keep potential age errors due to dating uncertainties in mind, the direct comparison shows a good match between both records for the entire Holocene. Major peaks in the Early to Mid-Holocene Sidi Ali $\delta^{18}O$ curve coincide with maxima in the ice rafted debris record (orange bars in Fig. 2). This is particularly evident at 11.4, 10.2, 9.3, 8.2 and 6.0 cal ka BP. As shown in Fig. 2b, these prominent peaks correspond with Bond events 8 to 4 (Bond et al.,

1997, 2001). The 25-point running correlation between both records is predominantly positive at those times (Fig. 2h).

In contrast, there is a noticeable negative correlation between Western Mediterranean winter rain minima and the ice rafted debris record during the Late Holocene (Fig. 2h). Here, low $\delta^{18}O$ values coincide with peaks in the ice rafted debris record

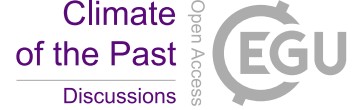

(blue bars in Fig. 2). Major troughs in the Sidi Ali $\delta^{18}$O curve at 4.2, 2.7, 1.2 and 0.5 cal ka BP concur with Bond events 3 to 0 (Fig. 2b). Hence, the compilation of our $\delta^{18}$O curve with the ice rafted debris record reveals **a hydro-climatic shift at ~5 cal ka BP** with multi-centennial intervals of Western Mediterranean winter rain minima and North Atlantic cooling during the Early and first half of the Mid-Holocene and opposite phases of winter rain maxima and North Atlantic cooling during the last

5,000 years.

*Evidence for a '4.2 ka Climatic Event' in the Western Mediterranean?*

Our manuscript is part of the 'Climate of the Past' special issue that deals about the '4.2 ka Climatic Event' and its probable global appearance. According to Weiss (2016), there is evidence for a 4.2-3.9 ka megadrought across the Mediterranean and

Western Asia that led to collapses of Early Bronze Age societies. The '4.2 ka Climatic Event' might correspond with North Atlantic Bond event 3 and there exists an ongoing debate in the scientific community about the global extent of a cold, dry and dusty multi-centennial event at that time. Central European palaeo-climatic archives, like well-dated Spannagel Cave speleothems in the Central Alps provide evidence for a cold and winter-dry climate around 4.2 ka (Mangini et al., 2007; Fohlmeister et al., 2012). Furthermore, in Central and Southern Italy many speleothems indicate a cold and dry climate around

4.2 ka (Di Rita and Magri, 2009; Margaritelli et al., 2016; Zanchetta et al., 2016). However, Western Mediterranean palaeo-environmental archives do not show uniform climatic patterns around 4.2 ka, although multiple studies report an arid interval at that time: in North-eastern Spain a prominent speleothem record indicates cold and dry conditions around 4.2 ka. According to Smith et al. (2016), this noticeable aridity interval is synchronous with large-scale North Atlantic cooling and an indicator for extending the spatial influence of the above mentioned 4.2 ka megadrought to the Western Mediterranean, or indeed into

the Atlantic sector of the Iberian Peninsula. In southern Spain, another speleothem record reveals a micro-hiatus at 4.16 ka that might correspond with the 4.2 ka Climatic Event (Walczak et al., 2015). These findings are supported by a pollen record from the Doñana National Park in south-western Spain that indicates a multi-centennial aridification trend centred at 4.0 cal ka BP (Jiménez-Moreno et al., 2015). Furthermore, a speleothem record from Gueldaman Cave in Northern Algeria reveals a multi-centennial dry phase in Western Mediterranean North Africa that started around 4.4 ka, and was synchronous with

abandonment of the cave (Ruan et al., 2016). However, $\delta^{18}$O and $\delta^{13}$C records of an adjacent speleothem at Gueldaman Cave do not show the same pattern and speleothem hydrochemistry might reflect also local factors. In contrast to the authors above, Cruz et al. (2015) assume a centennial-scale wet period at 4.2 ka from the Kaite Cave stalagmite record in Cantabrian Mountains of northern Spain.

According to the multi-proxy interpretation of the Sidi Ali $\delta^{18}$O (Fig. 2d), *Cedrus* (Fig. 2f) and microcharcoal (Fig. 2g) records, a centennial-scale interval of cool and wet conditions (pale blue bar in Fig. 2) represents the '4.2 ka Climatic Event' in the Middle Atlas. We point out that the Sidi Ali $^{14}$C age model of the 4.2 cal ka BP core section is based on two terrestrial plant residues (Fletcher et al., 2017) excluding any potential age uncertainties due to hard water effects. The correlation of the Sidi Ali records with the North Atlantic ice rafted debris record (Fig. 2c and 2d) shows that the local cool and wet interval at 4.2



cal ka BP is in-phase with cool conditions in the large-scale North Atlantic realm. Overall, this cool and wet interval fits into the in-phase hydro-climatic alternation of 'cool and wet' and 'warm and dry' conditions during the last 5,000 years in the Western Mediterranean (pale blue bars in Fig. 2). Therefore, the Sidi Ali record shows no dry event, respectively no out-of-phase climatic anomaly but increased humidity at 4.2 cal ka BP simultaneous with North Atlantic cooling.

**4.3 Drivers of the Holocene hydro-climate in the North Atlantic-Western Mediterranean region**

*Forcing mechanisms of Early Holocene Bond events and winter rain minima*

During the Early Holocene millennial-scale peaks in Sidi Ali winter rain minima parallel with North Atlantic Bond events 8 to 4 (orange bars in Fig. 2). Sidi Ali winter rain minima correspond with pollen-derived dry events in the Western Mediterranean lowlands (Fletcher et al., 2013), indicating a noticeable teleconnection between Western Mediterranean decreases in rainfall and North Atlantic cooling. Here, cooling over the North Atlantic was probably associated with a northward shift of Atlantic cyclone trajectories, leading to increased drought in the Western Mediterranean and Northern Africa (Zielhofer et al., 2017a). According to Bond et al. (2001) and Fletcher et al. (2013), North Atlantic cooling episodes, respectively ice rafted debris events result from millennial-scale weakening of the Atlantic Meridional Overturning Circulation (AMOC). Two of these 'cold relapses' (Wanner et al., 2011) correspond with prominent freshwater outbursts from the Laurentide ice sheet at 9.3 and 8.2 cal ka BP (Alley and Ágústsdóttir, 2005; Fleitmann et al., 2008), indicating evidence for an AMOC pattern during the deglaciation (Fletcher et al., 2013; Wassenburg et al., 2016) that is comparable with glacial conditions (Rahmsdorf, 2002; Moreno et al., 2005). The Early Holocene periodicity of 900 to 1,000 years in North Atlantic temperature changes and Western Mediterranean humidity is a widespread phenomenon in other palaeo-climatic records from the North Atlantic realm (Zhao et al., 2010; Cléroux et al., 2012; Fletcher et al., 2013) providing a significant coherence with the Eddy frequency band of total solar irradiance (Fig. 2a, Steinhilber et al., 2009).

*Forcing mechanisms of Late Holocene Bond events and winter rain maxima*

Atlantic winter cyclones and Western Mediterranean lows during spring control the present rainfall regime at Lake Sidi Ali in the Middle Atlas (Knippertz et al., 2003). Especially during the winter season, cool and wet air masses of the North Atlantic westerly circulation dominate the present hydro-climate in the Western Mediterranean basin (Born et al. 2010). Currently, the NAO significantly affects the amount of winter rainfall in the Western Mediterranean basin with increases in winter rainfall under negative NAO indices (Hurrell et al. 2003).

Likewise, there are multiple indications that the NAO represents a major forcing mechanism for past hydro-climatic changes in the Western Mediterranean. Both instrumental (Dünkeloh and Jacobeit 2003; Deininger et al. 2017) and also Late Holocene data (Magny et al. 2003; Baker et al., 2015; Corella et al., 2016; Wassenburg et al. 2016; Di Rita et al. 2018a, 2018b) provide evidence for spatio-temporal coherency in European precipitation pattern. Here, negative NAO indices correspond with increased effective winter rainfall in the Western Mediterranean and with decreased humidity in the Southern Mediterranean




and in Scandinavia. Following multiple authors (Trouet et al., 2009; Wassenburg et al., 2013), one of the most prominent negative NAO stages during the last 1,000 years occurred during the Little Ice Age. The Sidi Ali $\delta^{18}$O winter rain curve (Fig. 2d) shows similarities with a lake sediment record from southwestern Greenland (Olsen et al., 2012) that represents a NAO reconstruction over the past 5,200 years. Noticeable negative NAO stages of the Olsen record around 3.3, 2.7, 2.1 cal ka BP

and during the Little Ice Age correspond with winter rain peaks at Sidi Ali.

In this context, Late Holocene North Atlantic cooling and associated Western Mediterranean cooling and winter rain maxima (blue bars in Fig. 2) might reflect coupled atmosphere-ocean variability including subtropical gyre strength changes (Morley et al., 2011) that are paced by solar minima (Moffa-Sánchez et al., 2014). Here, Western Mediterranean winter rain maxima

coincide with multiple centennial-scale solar minima during the Late Holocene (Fig. 2a). This might be comparable with present NAO pattern that features primarily negative NAO indices during reduced solar irradiance (Matthes, 2011).

However, multi-centennial-scale shifts in Western Mediterranean hydro-climate and North Atlantic hydrography also show spatial differences that do not correspond with current NAO pattern: the International Ice Patrol's counts of icebergs crossing

48 ° N in southern direction are noticeably increased during positive indices of the NAO (Andrews 2000, USCG 2016). In this context, present iceberg variability is predominantly caused by fluctuation in Greenland ice sheet calving discharge rather than open ocean iceberg melting (Bigg et al. 2014). This does not correspond with the pairing of Bond's maxima in ice rafted debris and Sidi Ali winter rain maxima that would reflect negative NAO-like indices during the Late Holocene. Furthermore, Late Holocene ice rafted debris records from multiple North Atlantic marine cores (Bond et al., 2001) reveal synchronous iceberg

advances off Newfoundland, off Ireland and off Iceland. Bond's comparison with secondary palaeo-climatic records from the North Atlantic realm indicates that multi-centennial ice rafted debris events correspond with cooling phases in the entire region. This is not in accordance with the typical negative-NAO temperature pattern that shows sub-regional temperature increases in the subpolar North Atlantic (Bond et al., 2001). Spatially synchronous events of Holocene ice rafted debris are more typical for a reduced North Atlantic Deep Water formation (Moffa-Sánchez and Hall, 2018). In this context, the paleoceanographic

evidence for large-scale synchronous Holocene cooling events in the subpolar North Atlantic was recently verified by modelling results (Liu et al., 2017): a reduction of the AMOC corresponds with a widespread cooling over the northern North Atlantic and a noticeable sea ice expansion over the Greenland-Iceland-Norwegian seas.

In summary, the following conclusions for Late Holocene Bond events and Western Mediterranean winter rain maxima result:

the Late Holocene coincidence of Sidi Ali $\delta^{18}$O winter rain maxima and ice rafted debris events does not show strict spatial pattern and mechanisms of the present NAO. Rather, this centennial-scale pattern seems to be more typical for long-term AMOC variability with predominantly southward shifted westerlies and synchronous iceberg advances during intervals of reduced AMOC (Deininger et al., 2017). Here, major Late Holocene cooling events and Western Mediterranean winter rain maxima might correspond with centennial-scale solar minima (Fig. 2a; Steinhilber et al. 2009). Therefore, available 'NAO'





reconstructions (Trouet et al. 2009; Olsen et al. 2012; Wassenburg et al. 2016) might reflect a more complex set of forcing mechanisms (ice rafting, AMOC, solar forcing, NAO) influencing decadal to multi-centennial-scale changes in the North Atlantic hydro-climate during the past. In this context, our improved Sidi Ali $\delta^{18}$O winter rain record does not represent a strict NAO reconstruction but a hydro-climatic response of multi-centennial to millennial shifts in North Atlantic hydrography.

*Mid-Holocene hydro-climatic shift in the Western Mediterranean region*

Overall, the noticeable match between the Sidi Ali $\delta^{18}$O and the ice rafted debris records indicates a Holocene teleconnection between the subpolar North Atlantic and the Western Mediterranean hydro-climate but with a noticeable change in large-scale ocean-atmosphere coupled climatic mechanisms at ~5 cal ka BP. In contrast, some palaeoclimatic studies from the east of the

Iberian Peninsula (Pélachs et al. 2011; Smith et al., 2016) but also from the Alpine region (Mangini et al., 2007, Fohlmeister et al., 2012) postulate with reference to Bond's ice rafted debris record consistent oceanic-atmospheric interactions between the subpolar North Atlantic and Western Europe for the entire Holocene. However, high-resolution speleothem records from southern Spain (Walczak et al., 2015) and the Middle Atlas (Wassenburg et al. 2016), Tunisian alluvial records (Zielhofer and Faust, 2008) and an Alboran Sea pollen record (Fletcher et al. 2013) provide indications for a large-scale hydro-climatic shift

in the North Atlantic-Western Mediterranean region during the Mid-Holocene. This Mid-Holocene shift in Western Mediterranean hydro-climate is visible in significant frequency changes in humidity at multi-centennial time-scales but also at orbital scale. There is evidence for Early Holocene humidity and Late Holocene aridity in Mediterranean Morocco (Ibouhouten et al., 2010; Limondin-Lozouet et al., 2013), in Mediterranean domains of Algeria, Tunisia and Libya (Zielhofer et al., 2004; Bosmans et al., 2015; Wu et al. 2017) and in the Levant (Migowski et al., 2006, Zielhofer et al., 2018). Increased Early

Holocene rainfall in the Mediterranean basin corresponds with the African Humid Period in the North African monsoon domain (Bosmans et al., 2015; Shanahan et al., 2015) and reduced Saharan dust supply (Ehrmann et al., 2017; Zielhofer et al. 2017b). The Mid-Holocene southward shift of the ITCZ corresponds with a weakening and northward shift of the Atlantic winter storm tracks (Black et al., 2011; Kutzbach et al., 2014) and led to enduring drier winters in the Mediterranean basin during the Late Holocene.


**5 Conclusions**

Lake Sidi Ali is situated in the subhumid Middle Atlas Mountains of Northeast Morocco. Currently, the local hydro-climate is under strong influence of the NAO that provides enhanced effective rainfall under negative NAO indices. Previous palaeolimnological and -climatological studies indicate that the Middle Atlas represents a key region for Holocene hydro-

climatic variability in the Western Mediterranean.

In this study, we present an improved Holocene $\delta^{18}$O record of Sidi Ali ostracod shell material to enhance the chronological resolution of a previous record from the same core. The new data set provides a mean chronological resolution of 71.4 years.

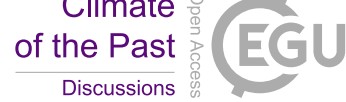

Peaks in the Sidi Ali δ¹⁸O record represent centennial to millennial-scale intervals of Western Mediterranean winter rain minima. This interpretation is based on a multi-proxy approach that includes *Cedrus* pollen and microcharcoal counts from the same core stratigraphy.

The coupling of the Sidi Ali δ¹⁸O record with the stacked ice rafted debris record (Bond et al. 2001) from the subpolar North Atlantic indicates a positive correlation during the Early Holocene and opposite pattern during the Late Holocene. Early Holocene Bond events, respectively North Atlantic cooling, parallel with cool and arid conditions in the Western Mediterranean, whereas during the last 5,000 years Bond events correspond with cool and wet hydro-climates. In the Early Holocene at least two Bond events at 9.3 and 8.2 cal ka BP coincide with prominent freshwater outbursts from the Laurentide

ice sheet.

Centennial-scale hydro-climatic anomalies show similarities with NAO pattern during the Late Holocene. However, our Sidi Ali δ¹⁸O record does not represent a strict NAO reconstruction but rather a hydro-climatic response of multi-centennial shifts in North Atlantic hydrography. Here, solar minima, iceberg advances, subtropical gyre strength changes and a reduced AMOC

are presented as major drivers of a coupled North Atlantic ocean-atmosphere system with multi-centennial intervals of Western Mediterranean winter rain maxima during the last 5,000 years.

Focusing on the '4.2 ka Climatic Event' that is a major subject of this 'Climate of the Past' special issue, our data show a cool and wet interval around 4.2 cal ka BP in the Western Mediterranean. This is overall in-phase with centennial-scale climatic

shifts from 'cool and wet' towards 'warm and dry' hydro-climates during the last 5,000 years.

**Data availability:** The newly conducted Sidi Ali oxygen stable isotope record will be provided in open access mode at https://www.researchgate.net/ after the final acceptance of the manuscript. In case of scientific use of the data, the citation of the primary source is obligatory: Zielhofer, C., Köhler, A., Mischke, S., Krüger, S., Benkaddour, A., Mikdad, A. and Fletcher,

W.J.: Hydro-climatic consequences of Holocene North Atlantic ice rafting events (Bond events) in the Western Mediterranean. Clim. Past.

**Supplement link:** Figure S1. Bayesian age model of the Sidi Ali core (Fletcher et al., 2017): the black dots show conventional ¹⁴C ages [BP]. The light greyish curves show ¹⁴C calibrated ages (prior). The dark greyish curves show modelled age

distributions (posterior). The light purple area show the 95% probability distribution. Calibration (2 sigma) of the conventional radiocarbon ages was performed using intcal13.14c.



**Author contribution:** CZ and WF designed the study; CZ and AK wrote a first draft of the manuscript; AK performed the statistical analyses and undertook the ostracod sampling; SM supervised the ostracod sampling procedure; CZ, AK, SM, AB, AM and WF contributed to discussion and revision of the manuscript.

**Competing interests:** The authors declare that they have no conflict of interest.

**Special issue statement:** The authors submit this contribution to the Climate of the Past Special Issue ""The 4.2ka BP Climatic Event". The manuscript focuses on a newly conducted Holocene oxygen stable isotope record of hydro-climatic variability in the Western Mediterranean that includes a multi-proxy interpretation about a potential hydro-climatic anomaly around the
4.2ka BP Climatic Event.

**Acknowledgements:** Christoph Zielhofer, Steffen Mischke and William Fletcher as principal investigators thank the German Research Foundation (DFG, ZI 721/9-1), the Federal Ministry of Education and Research (BMBF, 01DH17020) and the Natural Environment Research Council (New Investigator Award to W Fletcher, NE/K000608/1, and NERC RCF dating
awards, 1765.1013 and 1809.0414) for generous funding of the fieldwork and lab analyses. The authors are grateful to the Institut National des Sciences de l'Archéologie et du Patrimoine (INSAP, Rabat), the Centre National d'Hydrobiologie et de Pisciculture (CNHP, Azrou) and to the Caidad d'Azrou for helpful support in field and in preparing the expedition. Stefan Krüger and Thomas Brachert (Institute for Geophysics and Geology, Leipzig University) are highly acknowledged for conducting oxygen stable isotope data.

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
