# Peer review of "Western Mediterranean hydro-climatic consequences of Holocene iceberg advances (Bond events)"

_Climate of the Past, 2018_

## Referee Comment (RC1) · Anonymous Referee #1 · 16 Sep 2018

Review of "Western Mediterranean hydro-climatic consequences of Holocene iceberg advances (Bond events)" by Zielhofer et al.

The study presents an improved version of a published Holocene ostracod $\delta18O$ dataset from Lake Sidi Ali from the Middle Atlas in Morocco. The authors relate pronounced shifts in their record to the well-known Bond-events. The authors suggest that the study site's hydroclimate response to Atlantic cooling during Bond events changes from drier winters during the Early Holocene to wet winters during the Late Holocene. The paper is very well written, and I like the discussion on the Mid-Holocene climate shift, but I'm not convinced yet about the interpretation of the ostracod $\delta18O$, and not

all correlations with the Bond events.

Concerns: 1) The authors have published the interpretation of the $\delta$18O record in Zielhofer et al. (2017). They interpret the $\delta$18O record as a proxy for winter precipitation, which is based on a multi-proxy approach with a charcoal, and cedar pollen abundance records. Although, I can follow their line of arguments that cedar trees need enough moisture and the charcoal record may represent fire activity, I don't see a clear correlation between the charcoal record and the cedar pollen abundance. It is the coherence between these proxies that led the authors to the interpretation that the $\delta$18O represents winter precipitation. If this coherence is really there, then I would like to see a correlation matrix with significance levels between the cedrus pollen, charcoal, and ostracod $\delta$18O. This should be done for different timeslices, or perhaps with running correlations as done for the comparison with the Bond record. But in any case significance levels should be indicated.

2) The authors clearly state that lake Sidi Ali is a closed basin lake where the Precipitation - Evaporation balance (P-E) plays an essential role in controlling the oxygen isotope composition. This is evident from the highly elevated present day $\delta$18O values of the water that range from 0 to +4 ‰ whereas the surrounding karst springs and streams range from -6 to -9 ‰ (Zielhofer et al., 2017). The lake shows a huge range in surface area varying from 2 to 2.8 km2 (Zielhofer et al., 2017) due to varying P-E balance on interannual / decadal timescales. This is extremely likely visible in the $\delta$18O of the water. This can be controlled by both evaporation during the dry season, and by replenishment during the winter season, but not only through winter precipitation.

3) In order to show that the ostracod $\delta$18O variability represents the $\delta$18O of the water the authors calculate the theoretical calcite $\delta$18O values based on the present-day water $\delta$18O and the isotope fractionation factor from Friedman and O'Neill (1977). Why not using the much more recent isotope fractionation factor from Kim and O'Neill (1997)? Please show a range of possible temperatures that can be calculated taking into account different isotope fractionation factors.

4) One aspect that has not been discussed is the role of changing water temperatures on ostracod $\delta 18O$. Particularly during the Early Holocene where the authors argue that there are less cedar trees due to heat stress. There are four phases where there is a clear correlation with the $\delta 18O$ at 10.2, 8.2, 6.0, and 5.2 (I do not see a coherence at 7.3), why are these phases not interpreted as cooler summers? Cooler water temperatures may also result in heavier calcite $\delta 18O$, and could provide a different interpretation that is consistent with the cedar pollen abundance record. This may also be in line with "Atlantic cooling".

5) I do see a possible correlation with the HSG record from Bond et al. for the Early Holocene for the positive $\delta 18O$ peaks around 11.4, 10.2, 8.2. However, for the peaks at 9.3, 7.3, 6.7, 6.0, and 5.2 the variation in the $\delta 18O$ is either very small or the timing is not comparable to the Bond-events. The timing might be due to age-model uncertainties. But in its present form, I'm unable to assess whether the Bond events and the positive peaks in $\delta 18O$ are within error of the age model or not, because the age uncertainties are not indicated in Fig. 2. This is definitely a must.

6) The 25-point running correlation calculated between the $\delta 18O$ and the Bond record shows correlation that barely reach 0.3, is this significant? Can you draw a line that indicates the 95% confidence level? I'm aware that age model uncertainties should also be taken into account, so this can be discussed.

7) During the Late Holocene the authors try to link peaks at 4.6, 4.2, 3.2, 2.7 to peaks in HSG. I truly think that this is very hard to see, because the variation in $\delta 18O$ is very small. During the late Holocene these timings are also linked to cedrus pollen abundance peaks?? or troughs?? Without reading the text and by simply looking at the figure with the blue bars it is not possible for me to determine whether the authors think there is an increase or a decrease in the cedrus pollen abundance. Therefore, I find this unconvincing.

8) The paper shows no figure with a comparison with regional records to test their

interpretation of the ostracod $\delta18O$ record, for example the pollen record from MD95-2043 (Fletcher et al., 2013) should be included. Furthermore, if the authors are correct and their $\delta18O$ record represents winter precipitation, then a figure with a comparison with NAO records is necessary.

The discussion of the climate mechanisms in the paper is based on the interpretation of the ostracod $\delta18O$ record representing winter precipitation variability. However, in order to support the discussion, the authors need to show that the interpretation of the ostracod $\delta18O$ record is robust.

---

## Author Comment (AC1) · 13 Oct 2018

Dear Editor, Anonymous Referee #1 gave as important suggestions about our proxy data set. We will consider his comments in the revised version of our manuscript.

The following paragraphs are preliminary replies to the referee's comments:

Issue 1) "The authors have published the interpretation of the 18O record in Zielhofer et al. (2017). They interpret the 18O record as a proxy for winter precipitation, which is based on a multi-proxy approach with a charcoal, and cedar pollen abundance records. Although, I can follow their line of arguments that cedar trees need enough moisture

and the charcoal record may represent fire activity, I don't see a clear correlation between the charcoal record and the cedar pollen abundance."

In the first version of the CP manuscript, we applied a multi-proxy interpretation that is based on two scales. Orbital scale and multi-centennial to millennial scale.

At orbital scale $\delta$18O values increase from approx. +3 to +5 ‰ in the Early Holocene to values from +5 to +7 ‰ in the Late Holocene. As the most straightforward scenario for a subhumid, closed basin (Roberts et al., 2008) this implies a decrease of the precipitation/evaporation ratio with generally more arid conditions towards the Late Holocene. This corresponds with Sidi Ali diatom, TOC and carbonate records that indicate in average higher lake levels during the Early Holocene and lower levels at later stages (Zielhofer et al., 2017).

These findings seem to contradict the Cedrus pollen record from Lake Sidi Ali (Campbell et al., 2017) that shows a low or even missing occurrence of cedars during the Early Holocene, indicating reduced moisture availability at that time. However, reduced moisture availability for the cool-preferring cedar seems to be the result of enhanced summer heat during the Early Holocene. Due to their shallow roots, cedars are vulnerable to summer heat in contrast to the warm tolerant and deep-rooting evergreen oaks that dominate the Sidi Ali pollen record during the Early Holocene (Campbell et al., 2017). In this context, we attest summer temperature-driven drought stress and not winter precipitation as the limiting factor for the long-term trend of Holocene cedar occurrence in the Middle Atlas. This inference is in good agreement with the orbital-forced summer insolation maximum during the Early Holocene (Berger 1978) and the chironomid-based summer temperature reconstructions that indicate enhanced Mediterranean summer temperatures during the Early Holocene as well (Samartin et al., 2017).

Following our interpretation, at orbital scale our proxies show reduced winter precipitation (high $\delta$18O) and (summer) cool conditions (high amount of Cedrus pollen) during the Late Holocene and enhanced winter precipitation (low $\delta$18O) and enhanced summer heat (low Cedrus pollen amounts) during the Early Holocene. We think that this is currently the best interpretation but alternatives may exist. Furthermore, we worked out that enhanced $\delta18O$ values might be also the result of a specific origin and seasonality of the precipitation-bearing air masses. We assume that Late Holocene precipitation from springtime Mediterranean cyclones reveal higher $\delta18O$ values than Atlantic winter rains (Zielhofer et al. 2017).

However, we have to agree with Referee #1 that the orbital pattern between Cedrus and Sidi Ali $\delta18O$ is not clearly visible in the comparison of the Cedrus record with our $\delta18O$ record at multi-centennial and millennial time scales (here see Fig. 2 D and G) . In the revised manuscript, we will discuss this issue and will argue that the Cedrus record is influenced by summer heat stress and that summer heat might be predominantly forced by the subtropical high and not by North Atlantic air masses. This is visible in the "in phase" pattern between subtropical summer SST and our Cedrus record (Figure 1). We argue that reduced summer heat ("cooling at Sidi Ali") can be in phase with reduced winter rainfall at Sidi Ali (e.g. 8.2 and 10.2 ka) but that there are also indications for out-of-phase pattern (e.g. 1.2, 7.3, or 9.3 ka). Hence, this out-of-phase pattern might be influenced by different forcing mechanisms for summer cooling (sub-tropical high) and winter rain (North Atlantic winter cyclones). In the revised manuscript we will follow this line of argument that might explain the weak matching between Cedrus and $\delta18O$ at multi-centennial to millennial time scales. In the revised manuscript, we will argue that the Sidi Ali $\delta18O$ record is "in phase" with Bond's HSG record to support the idea of a teleconnection between Western Mediterranean winter precipitation and North Atlantic cooling. As Cedrus might reflect predominantly summer cooling we might not be able to detect North Atlantic (winter) cooling from our own record directly.

Due to uncertainties in age models (Sidi Ali vs. Bond record), we are not able to provide significant correlations between HSG and Sidi Ali $\delta18O$. Here, we argument more carefully now as suggested by referee #1. We avoid the term "correlation" and use the terms "in phase pattern" and "out of phase". Further, we apply lowpass filters (programme PAST) that reduce the centennial-scale variabilities of the proxy records (see Fig. 2). In the revised manuscript, therefore, we argue more carefully. The blue and orange bars in Fig. 2 indicate in phase pattern of the filtered HSG and $\delta$18O records. Further, we agree with referee #1 that the charcoal record is a difficult proxy for the interpretation of landscape dynamics at multi-centennial time scales. There might be no consistent matching between the charcoal and vegetation record. The charcoal can be dominated by individual fire events and these do not appear systematically linked to the fluctuations in the Cedrus pollen abundance. The charcoal is a complex proxy – influenced by climate but also by fuel availability, and the relative importance of these two factors seems to shift over the Holocene (Campbell et al. 2017). Therefore, we will not use the charcoal record for multi-centennial proxy interpretation in the revised manuscript.

Issue 2) "The authors clearly state that lake Sidi Ali is a closed basin lake where the Precipitation-Evaporation balance (P-E) plays an essential role in controlling the oxygen isotope composition. This is evident from the highly elevated present day 18O values of the water that range from 0 to +4 ‰ whereas the surrounding karst springs and streams range from -6 to -9 ‰ (Zielhofer et al., 2017). The lake shows a huge range in surface area varying from 2 to 2.8 km2 (Zielhofer et al., 2017) due to varying P-E balance on interannual/decadal timescales. This is extremely likely visible in the 18O of the water. This can be controlled by both evaporation during the dry season, and by replenishment during the winter season, but not only through winter precipitation."

We fully agree with Referee #1 that the lake is strongly affected by evaporation. This was clearly stated in the first version of the manuscript (page 7 line 15). Nevertheless, we tried to work out in our published manuscripts (Zielhofer et al. 2017 and Campbell et al. 2017) that the $\delta$18O record at Sidi Ali is a complex proxy and that the most convincing line of argument results in the variability of winter rains. We would like to mention here that we already worked out a convincing scenario for the early Holocene

millennial fluctuations (Zielhofer et al. 2017).

Issue 3) "In order to show that the ostracod 18O variability represents the 18O of the water the authors calculate the theoretical calcite 18O values based on the present-day water 18O and the isotope fractionation factor from Friedman and O'Neill (1977). Why not using the much more recent isotope fractionation factor from Kim and O'Neill (1997)? Please show a range of possible temperatures that can be calculated taking into account different isotope fractionation factors."

We followed this suggestion and used the equation by Kim and O'Neil (1997) to calculate $\delta$18O values for theoretical calcite based on measured water temperatures and the stable isotope composition of modern lake and spring and stream waters nearby. We corrected the resulting values in the text and changed the reference.

Further, we calculated additional $\delta$18O values (Table 1) for carbonate precipitated from Sidi Ali in equilibrium with host water at specific temperature scenarios and depths using the equation by Kim and O'Neil (1997) as suggested by the reviewer.

The lowest water temperature at the lake bottom in September 2012 was 7.7°C. Here, we have a $\delta$18O value (water, SMOW) of 1.21 ‰ for the depth of 30 m. Temperatures are quite stable around this depth. Using the Kim and O'Neil (1997) equation results in a $\delta$18O value (carbonate, VPDB) of 2.55 ‰ (The earlier used equation of Friedman and O'Neil (1977) had resulted in a value of 3.05 ‰ which is not so very different.)

The value of 2.55 ‰ is well within the range of data for the Holocene ostracod shells, which is -1.1 to 8.1 ‰ (min - max). In table 1, we combined measured and assumed temperature scenarios with today's $\delta$18O values (water, SMOW). For example, we had measured the highest $\delta$18O value with 2.58 ‰ (water, SMOW) at 5 m water depth in September 2012. Measured temperatures were highest in surface waters and were 19.6°C at maximum. With the exception of one $\delta$18O (carb) value for assumed 25°C that is slightly lower than our measured range for ostracod shells, all calculated theoretical $\delta$18O values (carbonate, VPDB) lie between 0.19 and 4.78 ‰ (Table 1) and are

in the range of our measured $\delta$18O values for Holocene ostracod shells at Sidi Ali.

However, we need to keep in mind that ostracod shells are not precipitated in isotopic equilibrium but often show a vital offset of 1 to 2 ‰ due to the metabolism of the animals (von Grafenstein et al., 1999). Therefore, $\delta$18O values of ostracod calcite are usually 1 to 2 ‰ higher than the assumed values for inorganic calcite. With this in mind, all calculated values in table 1 are within the range of our Holocene ostracod shell data and all these combinations are realistic and cannot be ruled out. The wider range of $\delta$18O from ostracod shells indicates that Holocene lake water $\delta$18O was sometimes higher and sometimes lower than today's $\delta$18O (water) values.

Further, the significantly wider range of Holocene $\delta$18O values from ostracods (-1.1 to 8.1 ‰ shows that solely water temperature changes cannot explain past $\delta$18O variability but changes in the precipitation/evaporation ratio must be considered as well.

Issue 4) "One aspect that has not been discussed is the role of changing water temperatures on ostracod 18O. Particularly during the Early Holocene where the authors argue that there are less cedar trees due to heat stress. There are four phases where there is a clear correlation with the 18O at 10.2, 8.2, 6.0, and 5.2 (I do not see a coherence at 7.3), why are these phases not interpreted as cooler summers? Cooler water temperatures may also result in heavier calcite 18O, and could provide a different interpretation that is consistent with the cedar pollen abundance record. This may also be in line with 'Atlantic cooling'".

Generally, this is exactly what we said. Increased Cedrus pollen indicate cooler summers (Page 8 line 10). However, multi-centennial changes in our $\delta$18O signal are quite large (more than 2 ‰ e.g. 8.2 ka) and the effect of temperature-dependent stable isotope fractionation during the formation of carbonate in water is not large enough for explaining these large changes. The core location is quite deep, and non-marine ostracods are (with very few exceptions) all benthic. Temperatures in modern Sidi Ali approach 8°C beneath the thermocline at 10-14 m. If temperatures were even as

low as 4°C (surely not colder at lake floor because of the density maximum of water at 4°C), $\delta$18O values could have been not higher than ca. 1 ‰ (see table 1) due to the temperature change. Early Holocene $\delta$18O fluctuations are often much larger in the record and water temperature alone was surely not driving these fluctuations. Even the slighter variabilities of Late Holocene $\delta$18O values cannot be inferred from temperature-dependent stable isotope fractionation because the Sidi Ali curve shows lower values during Atlantic cooling.

Issue 5) "I do see a possible correlation with the HSG record from Bond et al. for the Early Holocene for the positive 18O peaks around 11.4, 10.2, 8.2. However, for the peaks at 9.3, 7.3, 6.7, 6.0, and 5.2 the variation in the 18O is either very small or the timing is not comparable to the Bond-events. The timing might be due to age-model uncertainties. But in its present form, I'm unable to assess whether the Bond events and the positive peaks in 18O are within error of the age model or not, because the age uncertainties are not indicated in Fig. 2. This is definitely a must."

The age model and age uncertainties were submitted as supplementary online material in the first version of the manuscript. However, we add the error bars in the newly compiled figure (see Fig. 2). Further, we add lowpass filters for a better visualisation of "in phase" and "anti-phase" pattern between HSG and $\delta$18O. The filtered records (500yr lowpass filter) show a good match in multi-centennial variability.

Issue 6) "The 25-point running correlation calculated between the 18O and the Bond record shows correlation that barely reach 0.3, is this significant? Can you draw a line that indicates the 95% confidence level? I'm aware that age model uncertainties should also be taken into account, so this can be discussed."

We checked significance levels. Attained values above 0.4 and below approx. -0.4 are significant (95% confidence level). However, we will remove the 25-point correlation in the revised version of the manuscript and will argue more carefully ("in phase" vs. "out of phase").

Issue 7) "During the Late Holocene the authors try to link peaks at 4.6, 4.2, 3.2, 2.7 to peaks in HSG. I truly think that this is very hard to see, because the variation in 18O is very small."

We add lowpass filters for a better visualisation of in phase and anti-phase pattern between $\delta$18O and HSG records (see Fig. 2). The filtered records might provide a better visualisation.

Issue 8) "The paper shows no figure with a comparison with regional records to test their interpretation of the ostracod 18O record, for example the pollen record from MD95-2043 (Fletcher et al., 2013) should be included. Furthermore, if the authors are correct and their 18O record represents winter precipitation, then a figure with a comparison with NAO records is necessary."

We add a NAO record in the attached figure (Fig. 2 H; Olsen et al., 2012). However, the comparison is limited due to the high variability/higher resolution of the NAO record and age uncertainties. However, both records show similar pattern during the LIA and MCA, for example.

We have doubts about the value of showing MD95-2043 pollen record for comparison. The aim of this paper is to highlight a shift in phasing between $\delta$18O and the Bond record, and to suggest an explanation for that. For example, during the early Holocene (high summer orbital insolation, residual ice sheets), the Bond Events were associated with strong latitudinal temperature gradient and intensification of the westerly flow and weak penetration of winter rains into the W Mediterranean. During the Late Holocene (weak summer insolation, no ice sheets, modern ocean configuration), the Bond events may be more associated with ocean current changes around the dynamics of the ocean gyres and a similar-to-present linking of cold subpolar Atlantic & NAO-like negative pattern leading to increased rainfall. The MD95-2043 shows some similarities for the early Holocene but shows a slow changing millennial behaviour for the Late Holocene that does not really help support or refute the ideas about centennial variability at Sidi

Ali.

Kind regards Christoph Zielhofer, William Fletcher and Steffen Mischke

Table 1. Calculation of theoretical $\delta$18O values for carbonates precipitated from Lake Sidi Ali in equilibrium with host water at specific temperatures using the equation by Kim and O'Neil (1997).

Figure 1: Sidi Ali Cedrus record, Sidi Ali $\delta$18O and SST Hole 658C (1000 yr lowpass filter). The Cedrus record might show a good match with the subtropical SST record but not with winter rain (Sidi Ali $\delta$18O).

[revised manuscript text omitted]

---

## Author Comment (AC2) · 8 Nov 2018

Dear Editor, Anonymous Referee #1 [Review, 16th September 2018] suggested comparing our Sidi Ali $\delta$18O record with a NAO record to support the hypothesis that the Sidi Ali $\delta$18O signal represents a proxy for winter precipitation. We added in our first reply [Reply letter, 13th October 2018] the NAO record by Olsen et al. (2012) that covers the last 5.3 ka. However, due to the higher resolution of the NAO record a direct comparison between the NAO (Olsen et al., 2012) record and our Sidi Ali $\delta$18O record seems to be difficult. In the current reply letter, we added a 500 yr lowpass filter for the NAO record to facilitate a direct comparison (Fig. 2h). Although a direct comparison

between the filtered records (NAO and Sidi $\delta$18O, Fig. 2d and 2h) remains challenging due to the different resolution and scattering of the original datasets, similarities might be noticeable:

a) Both filtered records (NAO and Sidi Ali $\delta$18O) reveal troughs around 4.2, 3.3, and 2.7 cal ka BP that can be interpreted as increases in south-western Mediterranean winter precipitation. A generally cool and humid interval that is probably in phase with negative-NAO like pattern in the south-western and north-central Mediterranean basin around 4.2 cal ka BP (Bond event 3) is supported by a currently published manuscript by Di Rita and Magri (2018, this issue).

b) Both filtered records show a general transition from a more winter arid Medieval Climatic Anomaly (MCA) to a more winter humid Little Ice Age (LIA). This is in agreement with available NAO records from Mediterranean North Africa (Trouet et al. 2009, Wassenburg et al. 2013).

Regarding the first half of the Holocene, a currently published manuscript about the prominent Padul record in SW Spain (Ramos-Romána et al., 2018) supports our suggestion for Early to Mid-Holocene millennial cyclic fluctuations (cf. Fig. 2d, Sidi Ali $\delta$18O) in the south-western Mediterranean hydro-climate.

Finally, we slightly add and correct the captions (see below) of new figures 1 and 2 that were already provided in our previous reply letter [cf. Reply letter, 13th October 2018].

Olive bars in Fig. 1 indicate that Cedrus peaks at Sidi Ali correspond with Holocene summer cooling intervals in the sub-tropical belt. Therefore, we assume that maxima in the occurrence of Cedrus pollen might be mainly in line with summer cooling and less with variation in winter precipitation. This is also indicated by the sometimes-weak coincidence between Sidi $\delta$18O and the Cedrus record at multi-centennial to millennial time scales (e.g. at 1.8 cal ka BP, Fig. 1). In contrast, the comparison of the filtered (500 yr lowpass) haematite-stained grain record from the North Atlantic (Bond et al., 2001) with the filtered (500 yr lowpass) Sidi Ali $\delta$18O record (this study) indicate

noticeable similarities. Red numbers and pale red bars in fig. 2 indicate North Atlantic cooling events and dry Western Mediterranean winters in the first half of the Holocene. However, there seems to be a significant hydro-climatic shift at around 5 cal ka BP in the Western Mediterranean basin. Blue numbers and pale blue bars in fig. 2 indicate North Atlantic cooling events and wet Western Mediterranean winters in the second half of the Holocene.

Kind regards Christoph Zielhofer, William Fletcher, Steffen Mischke et al.

Figure captions

Figure 1. Holocene sub-tropical summer temperature record versus Sidi Ali Cedrus record and Western Mediterranean (Sidi Ali) winter rain record: A) Improved Sidi Ali $\delta$18O record from closely related species Fabaeformiscandona sp. and Candona sp. (Zielhofer et al., 2017 and this study). The grey line represents the original data. The black line shows results of lowpass filter (1000 year) removing centennial to multi-centennial variability. Red numbers indicate major dry winter phases in the Western Mediterranean; B) Hallstatt cyclicity based on 10Be sunspot number reconstruction (secondary singular spectrum analysis component [SSA2], Usoskin et al. 2016); C) Sidi Ali Cedrus pollen record (Campbell et al. 2017). Olive numbers and pale olive bars indicate synchronous phases of summer cooling in the Middle Atlas (Sidi Ali) and reduced summer SST in the subtropical North Atlantic; D) Summer sea surface temperature (SST) at Hole 658C (deMenocal et al., 2000). The grey line represents the original data. The black line shows results of lowpass filter (1000 year) removing centennial to multi-centennial variability.

Figure 2. Holocene North Atlantic ice-rafted debris record versus Western Mediter-ranean (Sidi Ali) winter rain record: A) Total solar irradiance ($\Delta$TSI, Steinhilber et al., 2009); B) Holocene Bond events 0 to 8 derived from Bond et al. (1997, 2001); C) Ice-rafted debris (IRD) record based on hematite stained grains of stacked MC52 and VM29-191 cores from the subpolar North Atlantic (Bond et al. 2001), the black line

shows results of lowpass filter (500 year) removing centennial variability; D) Improved Sidi Ali $\delta$18O record from closely related species Fabaeformiscandona sp. and Candona sp. (Zielhofer et al., 2017 and this study). The grey line represents the original data. The black line shows results of lowpass filter (500 year) removing centennial variability. Blue/red numbers and pale blue/orange bars indicate North Atlantic cooling events and wet/dry winters in the Western Mediterranean; E) Modelled ages with 2 sigma ranges (Fletcher et al., 2017); F) Summer insolation (65°N, June, Berger, 1978) (note reversed axis); G) Sidi Ali pollen record (Campell et al., 2017) with 500 year lowpass filter; H) Palaeo-NAO record (Olsen et al., 2012) with 500 year lowpass filter.

References

Berger, A., 1978. Long-term variations of caloric insolation resulting from the earth's orbital elements. Quat. Res. 9, 139-167.

Bond, G., Showers, W., Cheseby, M., Lotti, R., Almasi, P., deMenocal, P., Priore, P., Cullen, H., Hajdas, I. and Bonani, G.: A pervasive millennial-scale cycle in North Atlantic Holocene and glacial climates. Science 278, 1257-1266, 1997.

Bond, G., Kromer, B., Beer, J., Muscheler, R., Evans, M.N., Showers, W., Hoffmann, S., Lotti-Bond, R., Hajdas, I. and Bonani, G.: Persistent Solar Influence on North Atlantic Climate during the Holocene. Science 294, 2130-2136, 2001.

Campbell, J.F.E., Fletcher, W.J., Joannin, S., Hughes, P., Rhanem, M. and Zielhofer, C.: Environmental drivers of Holocene forest development in the Middle Atlas, Morocco. Front. Ecol. Evol. 5, 113, doi: 10.3389/fevo.2017.00113, 2017.

deMenocal, P.B., Ortiz, J., Guilderson, T., Sarnthein, M., 2000. Coherent high- and low-latitude climate variability during the Holocene warm period. Science 288, 2198-2202.

Di Rita, F., Magri, D., 2018. The 4.2 ka BP event in the vegetation record of the central Mediterranean. Clim. Past Discuss., https://doi.org/10.5194/cp-2018-128 ("The 4.2 ka

[Figure]

Climatic Event" Special Issue).

Fletcher, W.J., Zielhofer, C., Mischke, S., Bryant, C., Xu, X., Fink, D., 2017. AMS radiocarbon dating of pollen concentrates in a karstic lake system. Quat. Geochronol. 39, 112-123.

Olsen, J., Anderson, N.J., Knudsen, M.F., 2012. Variability of the North Atlantic Oscillation over the past 5,200 years. Nat. Geosci. 5, 808-812.

Ramos-Romána, M.J., Jiménez-Moreno, G., Camuera, J., García-Alix, A., Scott Anderson, R., Jiménez-Espejo, F.J., Sachse, D., Toney, J.L., Carrión, J.S., Webster, C., Yanes, Y., 2018. Millennial-scale cyclical environment and climate variability during the Holocene in the western Mediterranean region deduced from a new multiproxy analysis from the Padul record (Sierra Nevada, Spain). Glob. Planet. Change 168, 35-53.

Trouet, V., Esper, J., Graham, N.E., Baker, A., Scourse, J.D., Frank, D.C., 2009. Persistent positive North Atlantic Oscillation mode dominated the Medieval climate anomaly. Science 324, 78-80.

Usoskin, I.G., Gallet, Y., Lopes, F., Kovaltsov, G.A., Hulot, G., 2016. Solar activity during the Holocene: the Hallstatt cycle and its consequence for grand minima and maxima, Astron. Astrophys., 587, A150, doi: 10.1051/0004-6361/201527295

Wassenburg, J.A., Immenhauser, A., Richter, D.K., Niedermayr, A., Riechelmann, S., Fietzke, J., Scholz, D., Jochum, K.P., Fohlmeister, J., Schröder-Ritzrau, A., Sabaoui, A., Riechelmann, D.F.C., Schneider, L., Esper, J., 2013. Moroccan speleothem and tree ring records suggest a variable positive state of the North Atlantic Oscillation during the Medieval Warm Period. Earth Planet. Sci. Lett. 375, 291-301.

Zielhofer, C., Fletcher, W.J., Mischke, S., De Batist, M., Campbell, J.F.E., Joannin, S., Tjallingii, R., El Hamouti, N., Junginger, A., Stele, A., Bussmann, J., Schneider, B., Lauer, T., Spitzer, K., Strumpler, M., Brachert, T. and Mikdad, A.: Atlantic forcing of Western Mediterranean winter rain minima during the last 12,000 years. Quat. Sci.

Rev. 157, 29-51, 2017.

[Figure]

[Figure]

**Fig. 1.** Holocene sub-tropical summer temperature record versus Sidi Ali Cedrus record and Western Mediterranean (Sidi Ali) winter rain record (see text file for more details)

[Figure]

**Fig. 2.** Holocene North Atlantic ice-rafted debris record versus Western Mediterranean (Sidi Ali) winter rain record (see text file for more details)

---

## Referee Comment (RC2) · Anonymous Referee #2 · 27 Dec 2018

Interactive comment on "Western Mediterranean hydro-climatic consequences of Holocene iceberg advances (Bond events)". Christoph Zielhofer et al.

Anonymous Reviewer 2

Introduction and methodology: This study leads off with a thoughtful introduction reviewing and analyzing the North Atlantic (NA) rafted debris record (Bond events) and makes a strong case for Mediterranean studies showing probable linkages of hydroclimate and the Bond event record. Studies identified and compared in this work are well summarized, represent a substantial range of Mediterranean sites, and their records compared to highlight regional variability of humidity and dryness, and initially, the authors emphasize caution in attributing these patterns (in response to Atlantic cooling events) to "forcing mechanisms, or chronological correlations". The introduction is bolstered by three well designed figures that present a broad to fine scale descriptions of the study area and place in context the North Atlantic Basin, regional climate patterns, and the coring site depicting the local landscape and vegetation.

The methodology was one of the strengths of this study, with the with addition of 82 new samples to a previously published $\delta$18O ostracod record for Lake Sidi Ali in the Middle Atlas range of Morocco. The new samples bring a total number of data points to 182 for 12.97 m record spanning the Early to Late Holocene (12K cal ybp), and almost doubling the 14C chronological sample resolution of the previous record from "~130 years to 71.4 years". This robust record is reinforced by 210Pb and 137Cs dating in the historic. The $\delta$18O data were further compared with pollen (Cedrus sp.), micro-charcoal, solar activity, solar insolation, as well as a running 25-point correlation between the Bond event IRD record and the Sidi Ali $\delta$18O record. Clear figures, stacked with a color overprint of Early to Late Holocene hydroclimate changes and Bond event intervals, strongly reinforce the authors thinking.

Hydroclimate: The authors characterize the overall pattern of the Sidi Ali Record in the Early Holocene with Atlantic cooling coupled with dry winters with higher summer temperatures producing drought stress limiting Cedrus. In addition, the early summer warm climate co-occurs with warm Atlantic winter rains, except during Bond events. The record is dramatically reversed, in this reviewer's opinion, for the Late Holocene beginning about 5K cal ybp where Atlantic cooling produces wet winters, in a hydroclimate of decreasing rainfall. Both Early and Late Holocene interpretations are supported from additional studies with TOC, diatom, and charcoal; and solar forcing, solar insolation, and chironomid data, respectively. This two-phase change in the Early and Late Holocene $\delta$18O record could be described as a marked low frequency, high amplitude signal that sharply decreases in amplitude after 5K cal ybp and the into the Mid-Late Holocene, and arguably begins to increase in amplitude and frequency from

$\sim$ 2K cal ybp into to the modern.

In addressing the $\delta$18O record, this reviewer suggests more description and/or insights from the authors would be helpful to interpret and the patterns of the signal with regards to amplitude and frequency, which this reader found dramatic. Possibly presenting these data with some type of signal-to-noise ratio analyses could be helpful. Additionally, this same approach could be beneficial comparing and evaluating both the pollen and charcoal data. Clearly, there is a wide range of pollen responses between the 10.2 and 7.2 Bond events, using the Early Holocene as an example. And while both responses are positive, they are clearly different in their absolute values, and appear dissimilar. I would find some characterization and analysis of this variability helpful. A similar argument can be made for the Late Holocene segment in the record, again, especially for the charcoal and pollen records. The Late Holocene charcoal signal depicts an increase in peak values of charcoal, and a variable higher frequency pattern of peaks. Some method for identifying fire events, either a threshold of signal to noise ratio, or a confidence interval set from smoothed baseline could potentially sharpen the fire event interpretation. Finally, trends in the pollen data, such as the slower rise of Cedrus before the Early/Late Holocene shift in $\delta$18O record, the change in the frequency of the pollen signal, and frequent occurrence of charcoal peaks beginning about 5K cal ybp also suggest there may be additional ecological factors influencing the Cedrus pollen response. Appreciated was the acknowledgment, that settlement history may have had an additional influence in the charcoal record, as is the case with many Holocene paleo-fire reconstructions.

4.2 Bond event: An additional strength of this study was the scholarship involved in the introduction and summary of a wide range of regional studies placing the results of the authors in context with the broader Mediterranean. Especially helpful, was the discussion of hydroclimate variability specific to other environmental reconstructions of the 4.2 period across the Mediterranean. The findings of this study showing a cool wet event at the 4.2 Bond event, was nicely contrasted with a number regional studies

mostly indicating the 4.2 as a period of dryness. Further, the thoroughness of the authors placing this study in the context of such a wide spectrum of studies throughout the entire paper, potentially could be improved by a table or matrix figure summarizing this study's results and the many citations included within, bringing readers less familiar with Holocene paleo-environments of the Mediterranean region, into the sphere of thinking of the authors.

Conclusions: This is a well written paper, and should be published with minor revisions. The amplification of the resolution of a previously published record, and subsequent interpretation of that record, is an important and detailed contribution to the understanding of paleo-hydroclimate dynamics of Morocco. In addition, the paper makes a significant case for further investigations of Holocene paleo-hydroclimate scenarios in a broader Mediterranean context, with the comparison of this record to numerous efforts, emphasizing "coherence with Bond events across the entire Holocene" for some areas, yet in contrast, other sites demonstrating a variable step change from a wetter Early Holocene to a Late-Holocene of aridity at 5K cal ybp.

Post script comments on the author response to Anonymous Reviewer 1: Figure, table, and comments submitted in both responses to Reviewer 1, greatly strengthened this submission. Especially helpful were 2 sigma age model panel, in addressing uncertainties, and the 500-year low pass filter on the pollen record and $\delta$18O reconstruction clarifying the Cedrus response.

Please also note the supplement to this comment:
https://www.clim-past-discuss.net/cp-2018-97/cp-2018-97-RC2-supplement.pdf

---

## Author Comment (AC3) · 23 Jan 2019

Dear Editor, Anonymous Referee #2 gave as important suggestions about our proxy data set. We will consider his comments in the revised version of our manuscript.

The following paragraphs provide replies to the referee#2's comments:

Ref#2: "Introduction and methodology: This study leads off with a thoughtful introduction reviewing and analyzing the North Atlantic (NA) rafted debris record (Bond events) and makes a strong case for Mediterranean studies showing probable linkages of hydroclimate and the Bond event record. Studies identified and compared in this work

are well summarized, represent a substantial range of Mediterranean sites, and their records compared to highlight regional variability of humidity and dryness, and initially, the authors emphasize caution in attributing these patterns (in response to Atlantic cooling events) to 'forcing mechanisms, or chronological correlations'".

We thank the ref#2 for the general positive comment.

Ref#2: "The introduction is bolstered by three well designed figures that present a broad to fine scale descriptions of the study area and place in context the North Atlantic Basin, regional climate patterns, and the coring site depicting the local landscape and vegetation."

We thank the ref#2 for the general positive comment.

Ref#2: "The methodology was one of the strengths of this study, with the with addition of 82 new samples to a previously published 18O ostracod record for Lake Sidi Ali in the Middle Atlas range of Morocco. The new samples bring a total number of data points to 182 for 19.63 m record spanning the Early to Late Holocene (12K cal ybp), and almost doubling the 14C chronological sample resolution of the previous record from "∼130 years to 71.4 years". This robust record is reinforced by 210Pb and 137Cs dating in the historic. The 18O data were further compared with pollen (Cedrus sp.), micro-charcoal, solar activity, solar insolation, as well as a running 25-point correlation between the Bond event IRD record and the Sidi Ali 18O record. Clear figures, stacked with a color overprint of Early to Late Holocene hydroclimate changes and Bond event intervals, strongly reinforce the authors thinking."

We thank the ref#2 for the general positive comment. Further, we considered comments of the ref#1 and integrated results of low-pass filtering in the figures. Additionally, we want to restructure the manuscript according to the issues of the ref#1 (cf. Zielhofer et al. 2018)

Ref#2: "Hydroclimate: The authors characterize the overall pattern of the Sidi Ali

Record in the Early Holocene with Atlantic cooling coupled with dry winters with higher summer temperatures producing drought stress limiting Cedrus. In addition, the early summer warm climate co-occurs with warm Atlantic winter rains, except during Bond events. The record is dramatically reversed, in this reviewer's opinion, for the Late Holocene beginning about 5K cal ybp where Atlantic cooling produces wet winters, in a hydroclimate of decreasing rainfall. Both Early and Late Holocene interpretations are supported from additional studies with TOC, diatom, and charcoal; and solar forcing, solar insolation, and chironomid data, respectively. This two-phase change in the Early and Late Holocene 18O record could be described as a marked low frequency, high amplitude signal that sharply decreases in amplitude after 5K cal ybp and the into the Mid-Late Holocene, and arguably begins to increase in amplitude and frequency from ∼2K cal ybp into to the modern."

We thank the ref#2 for the general positive comment.

Ref#2: "In addressing the 18O record, this reviewer suggests more description and/or insights from the authors would be helpful to interpret and the patterns of the signal with regards to amplitude and frequency, which this reader found dramatic. Possibly presenting these data with some type of signal-to-noise ratio analyses could be helpful."

Many thanks for this comment. Especially we will emphasise in the discussion chapter the change in amplitude of the 18O signal at ∼5 ka.

Ref#2: "Additionally, this same approach could be beneficial comparing and evaluating both the pollen and charcoal data. Clearly, there is a wide range of pollen responses between the 10.2 and 7.2 Bond events, using the Early Holocene as an example. And while both responses are positive, they are clearly different in their absolute values, and appear dissimilar. I would find some characterization and analysis of this variability helpful. A similar argument can be made for the Late Holocene segment in the record, again, especially for the charcoal and pollen records. The Late Holocene charcoal signal depicts an increase in peak values of charcoal, and a variable higher frequency

pattern of peaks. Some method for identifying fire events, either a threshold of signal to noise ratio, or a confidence interval set from smoothed baseline could potentially sharpen the fire event interpretation."

Many thanks. Actually, we do not plan to follow this suggestion in the overall revision of the manuscript because we restructured the manuscript following the issues of ref#1 and put the charcoal record out of the millennial-scale interpretation. Further, in the revised version the Cedrus pollen record is discussed as a probable proxy for summer temperature. We added a 1000 yr low-pass filter (Zielhofer et al. 2018) for a better illustration of potential coincidences between decreased subtropical summer heat (deMenocal et al. 2000) and Cedrus increases.

Ref#2: "Finally, trends in the pollen data, such as the slower rise of Cedrus before the Early/Late Holocene shift in $\delta18O$ record, the change in the frequency of the pollen signal, and frequent occurrence of charcoal peaks beginning about 5K cal ybp also suggest there may be additional ecological factors influencing the Cedrus pollen response. Appreciated was the acknowledgment, that settlement history may have had an additional influence in the charcoal record, as is the case with many Holocene paleo-fire reconstructions."

Many thanks. Yes, we add additional thoughts about the ecological factors influencing Cedrus response. Here, the bi-millennial frequency of the Cedrus record might be strong during the Early Holocene due to the generally higher impact of summer solar radiation at that time.

Ref#2: "4.2 Bond event: An additional strength of this study was the scholarship involved in the introduction and summary of a wide range of regional studies placing the results of the authors in context with the broader Mediterranean. Especially helpful, was the discussion of hydroclimate variability specific to other environmental reconstructions of the 4.2 period across the Mediterranean. The findings of this study showing a cool wet event at the 4.2 Bond event, was nicely contrasted with a number

regional studies mostly indicating the 4.2 as a period of dryness. Further, the thoroughness of the authors placing this study in the context of such a wide spectrum of studies throughout the entire paper, potentially could be improved by a table or matrix figure summarizing this study's results and the many citations included within, bringing readers less familiar with Holocene paleo-environments of the Mediterranean region, into the sphere of thinking of the authors."

We thank the ref#2 for the general positive comment and the helpful suggestion. Right, in the final version of the manuscript we will integrate a matrix figure summarizing major study's conclusions (Fig. 1).

Ref#2: "Conclusions: This is a well written paper, and should be published with minor revisions. The amplification of the resolution of a previously published record, and subsequent interpretation of that record, is an important and detailed contribution to the understanding of paleo-hydroclimate dynamics of Morocco. In addition, the paper makes a significant case for further investigations of Holocene paleo-hydroclimate scenarios in a broader Mediterranean context, with the comparison of this record to numerous efforts, emphasizing "coherence with Bond events across the entire Holocene" for some areas, yet in contrast, other sites demonstrating a variable step change from a wetter Early Holocene to a Late-Holocene of aridity at 5K cal ybp."

We thank the ref#2 for the general positive comment.

Ref#2: "Post-script comments on the author response to Anonymous Reviewer 1: Figure, table, and comments submitted in both responses to Reviewer 1, greatly strengthened this submission. Especially helpful were 2 sigma age model panel, in addressing uncertainties, and the 500-year low pass filter on the pollen record and 18O reconstruction clarifying the Cedrus response."

We thank the ref#2 for the general positive comment.

Kind regards Christoph Zielhofer and all co-authors

References

deMenocal, P.B., Ortiz, J., Guilderson, T. & Sarnthein, M. 2000b. Coherent High- and Low-Latitude Climate Variability During the Holocene Warm Period. Science 288, 2198-2202.

Zielhofer, C., Köhler, A., Mischke, S., Benkaddour, A., Mikdad, A., Fletcher, W.J., 2018. Interactive comment on "Western Mediterranean hydro-climatic consequences of Holocene iceberg advances (Bond events)" Clim. Past Discuss., https://doi.org/10.5194/cp-2018-97-RC2, 2018

[Figure]

[Figure]

**Fig. 1.** Lake Sidi Ali Holocene core: illustration of major conclusions

---

## Author Response (AR1)

Dear Raymond Bradley,

**Anonymous Referee #1** gave as important suggestions about our proxy data set. We considered his comments in the fully revised version of the manuscript.

 *"Concern: 1) The authors have published the interpretation of the 18O record in Zielhofer et al. (2017). They interpret the 18O record as a proxy for winter precipitation, which is based on a multi-proxy approach with a charcoal, and cedar pollen abundance records. Although, I can follow their line of arguments that cedar trees need enough moisture and the charcoal record may represent fire activity, I don't see a clear correlation between the charcoal record and the cedar pollen abundance."*

In the first version of the CP manuscript, we applied a multi-proxy interpretation that is based on two scales. Orbital scale and multi-centennial to millennial scale.

At orbital scale $\delta^{18}O$ values increase from approx. +3 to +5 ‰ in the Early Holocene to values from +5 to +7 ‰ in the Late Holocene. As the most straightforward scenario for a subhumid, closed basin (Roberts et al., 2008) this implies a decrease of the precipitation/evaporation ratio with generally more arid conditions towards the Late Holocene. This corresponds with Sidi Ali diatom, TOC and carbonate records that indicate in average higher lake levels during the Early Holocene and lower levels at later stages (Zielhofer et al., 2017).

These findings seem to contradict the *Cedrus* pollen record from Lake Sidi Ali (Campbell et al., 2017) that shows a low or even missing occurrence of cedars during the Early Holocene, indicating reduced moisture availability at that time. However, reduced moisture availability for the cool-preferring cedar seems to be the result of enhanced summer heat during the Early Holocene. Due to their shallow roots, cedars are vulnerable to summer heat in contrast to the warm tolerant and deep-rooting evergreen oaks that dominate the Sidi Ali pollen record during the Early Holocene (Campbell et al., 2017). In this context, we attest summer temperature-driven drought stress and not winter precipitation as the limiting factor for the long-term trend of Holocene cedar occurrence in the Middle Atlas. This inference is in good agreement with the orbital-forced summer insolation maximum during the Early Holocene (Berger 1978) and the chironomid-based summer temperature reconstructions that indicate enhanced Mediterranean summer temperatures during the Early Holocene as well (Samartin et al., 2017).

Following our interpretation, at orbital scale our proxies show reduced winter precipitation (high $\delta^{18}O$) and (summer) cool conditions (high amount of *Cedrus* pollen) during the Late Holocene and enhanced winter precipitation (low $\delta^{18}O$) and enhanced summer heat (low *Cedrus* pollen amounts) during the Early Holocene. We think that this is currently the best interpretation but alternatives may exist. Furthermore, we worked out that enhanced $\delta^{18}O$ values might be also the result of a specific origin and seasonality of the precipitation-bearing air masses. We assume that Late Holocene precipitation from springtime Mediterranean cyclones reveal higher $\delta^{18}O$ values than Atlantic winter rains (Zielhofer et al. 2017).

However, we have to agree with Referee #1 that the orbital pattern between *Cedrus* and Sidi Ali $\delta^{18}O$ is not clearly visible in the comparison of the *Cedrus* record with our $\delta^{18}O$ record at multi-centennial and millennial time scales. In the fully revised manuscript, we discuss this issue and argue that the *Cedrus* record is influenced by summer heat stress and that summer heat might be predominantly forced by the subtropical high and not by North Atlantic air masses. This is visible in the "in phase" pattern between subtropical summer SST and our *Cedrus* record (new Figure 2). We argue that reduced summer heat ("cooling at Sidi Ali") can be in phase with reduced winter rainfall at Sidi Ali (e.g. 8.2 and 10.2 ka) but that there are also indications for out-of-phase pattern (e.g. during the Late Holocene and also at 7.3 and 9.3 cal ka BP). Hence, this out-of-phase pattern might be influenced by

different forcing mechanisms for summer cooling (sub-tropical high) and winter rain (North Atlantic winter cyclones). In the fully revised manuscript, we follow this line of argument that might explain the weak matching between *Cedrus* and δ$^{18}$O at multi-centennial to millennial time scales. In the fully revised manuscript, we argue that the Sidi Ali δ$^{18}$O record is "in phase" with Bond's HSG record to support the idea of a teleconnection between Western Mediterranean winter precipitation and North Atlantic cooling. As *Cedrus* might reflect predominantly summer cooling we are not be able to detect North Atlantic (winter) cooling from our own record directly.

Due to uncertainties in age models (Sidi Ali vs. Bond record), we are not able to provide significant correlations between HSG and Sidi Ali δ$^{18}$O. Here, we argument more carefully now as suggested by referee #1. We avoid the term "correlation" and use the terms "in phase pattern" and "out of phase". Further, we apply lowpass filters (programme PAST) that reduce the centennial-scale variabilities of the proxy records (see new Figs. 2 and 3). In the fully revised manuscript, therefore, we argue more carefully. The blue and orange bars in Figure 3 indicate in phase pattern of the filtered HSG and δ$^{18}$O records.

Further, we agree with referee #1 that the charcoal record is a difficult proxy for the interpretation of landscape dynamics at multi-centennial time scales. There might be no consistent matching between the charcoal and vegetation record. The charcoal can be dominated by individual fire events and these do not appear systematically linked to the fluctuations in the Cedrus pollen abundance. The charcoal is a complex proxy – influenced by climate but also by fuel availability, and the relative importance of these two factors seems to shift over the Holocene (Campbell et al. 2017). Therefore, we will not use the charcoal record in the revised manuscript.

2) The authors clearly state that lake Sidi Ali is a closed basin lake where the Precipitation-Evaporation balance (P-E) plays an essential role in controlling the oxygen isotope composition. This is evident from the highly elevated present day 18O values of the water that range from 0 to +4 ‰ whereas the surrounding karst springs and streams range from -6 to -9 ‰ (Zielhofer et al., 2017). The lake shows a huge range in surface area varying from 2 to 2.8 km2 (Zielhofer et al., 2017) due to varying P-E balance on interannual/decadal timescales. This is extremely likely visible in the 18O of the water. This can be controlled by both evaporation during the dry season, and by replenishment during the winter season, but not only through winter precipitation.

We fully agree with Referee #1 that the lake is strongly affected by evaporation. This was clearly stated in the first version of the manuscript and is again stated in the fully revised version (page 8 line 5). Nevertheless, we tried to work out in our published manuscripts (Zielhofer et al. 2017 and Campbell et al. 2017) that the δ$^{18}$O record at Sidi Ali is a complex proxy and that the most convincing line of argument results in the variability of winter rains.

3) In order to show that the ostracod 18O variability represents the 18O of the water the authors calculate the theoretical calcite 18O values based on the present-day water 18O and the isotope fractionation factor from Friedman and O'Neill (1977). Why not using the much more recent isotope fractionation factor from Kim and O'Neill (1997)? Please show a range of possible temperatures that can be calculated taking into account different isotope fractionation factors.

We followed this suggestion and used the equation by Kim and O'Neil (1997) to calculate δ$^{18}$O values for theoretical calcite based on measured water temperatures and the stable isotope composition of modern lake and spring and stream waters nearby. We corrected the resulting values in the text and changed the reference.

Further, we calculated additional $\delta^{18}O$ values (Table 1) for carbonate precipitated from Sidi Ali in equilibrium with host water at specific temperature scenarios and depths using the equation by Kim and O'Neil (1997) as suggested by the reviewer.

The lowest water temperature at the lake bottom in September 2012 was 7.7°C. Here, we have a $\delta^{18}O$ value (water, SMOW) of 1.21 ‰ for the depth of 30 m. Temperatures are quite stable around this depth. Using the Kim and O'Neil (1997) equation results in a $\delta^{18}O$ value (carbonate, VPDB) of 2.55 ‰. (The earlier used equation of Friedman and O'Neil (1977) had resulted in a value of 3.05 ‰, which is not so very different.)

The value of 2.55 ‰ is well within the range of data for the Holocene ostracod shells, which is -1.1 to 8.1 ‰ (min - max). In table 1, we combined measured and assumed temperature scenarios with today's $\delta^{18}O$ values (water, SMOW). For example, we had measured the highest $\delta^{18}O$ value with 2.58 ‰ (water, SMOW) at 5 m water depth in September 2012. Measured temperatures were highest in surface waters and were 19.6°C at maximum. With the exception of one $\delta^{18}O$ (carb) value for assumed 25°C that is slightly lower than our measured range for ostracod shells, all calculated theoretical $\delta^{18}O$ values (carbonate, VPDB) lie between 0.19 and 4.78 ‰ (Table 1) and are in the range of our measured $\delta^{18}O$ values for Holocene ostracod shells at Sidi Ali.

However, we need to keep in mind that ostracod shells are not precipitated in isotopic equilibrium but often show a vital offset of 1 to 2 ‰ due to the metabolism of the animals (von Grafenstein et al., 1999). Therefore, $\delta^{18}O$ values of ostracod calcite are usually 1 to 2 ‰ higher than the assumed values for inorganic calcite. With this in mind, all calculated values in table 1 are within the range of our Holocene ostracod shell data and all these combinations are realistic and cannot be ruled out. The wider range of $\delta^{18}O$ from ostracod shells indicates that Holocene lake water $\delta^{18}O$ was sometimes higher and sometimes lower than today's $\delta^{18}O$ (water) values.

Further, the significantly wider range of Holocene $\delta^{18}O$ values from ostracods (-1.1 to 8.1 ‰) shows that solely water temperature changes cannot explain past $\delta^{18}O$ variability but changes in the precipitation/evaporation ratio must be considered as well. In the fully revised version we consider these issues and refer to Zielhofer et al. 2018b for full details.

Table 1. Calculation of theoretical $\delta^{18}O$ values for carbonates precipitated from Lake Sidi Ali in equilibrium with host water at specific temperatures using the equation by Kim and O'Neil (1997).

| Lake water depth | Temp. | $\delta^{18}O$ (water, SMOW) | Theoretical $\delta^{18}O$ (carb., VPDB) | Remarks |
|---|---|---|---|---|
| 30 m | 7.7 °C (Sep. 2012) | 1.21 ‰ (Sep. 2012) | 2.55 ‰ | Calculation for actually measured water temperature and $\delta^{18}O$ (water) at 30 m |
| 0 to 5 m | 19.6 °C (Sep. 2012) | 2.58 ‰ (Sep. 2012) | 1.30 ‰ | Calculation for maximum measured $\delta^{18}O$ (water) at 5 m depth and maximum surface water temperature (ca. 1 °C warmer than at 5 m) |
| 30 m | 4 °C | 1.21 ‰ (Sep. 2012) | 3.41 ‰ | Calculation for assumed coldest water during colder times and present bottom water $\delta^{18}O$ (water) |
| 30 m | 4 °C | 2.58 ‰ | 4.78 ‰ | Calculation for assumed coldest water during colder times and more evaporated water (similar to surface water) |
| 0 to 5 m | 25 °C | 1.21 ‰ | -1.18 ‰ | Calculation for assumed very warm (or shallow) conditions and present bottom water $\delta^{18}O$ (water) |

| 0 to 5 m | 25 °C | 2.58 ‰ | 0.19 ‰ | Calculation for assumed very warm (or shallow) conditions and today's maximum measured $\delta^{18}O$ (water) at 5 m depth |
|---|---|---|---|---|

4) One aspect that has not been discussed is the role of changing water temperatures on ostracod 18O. Particularly during the Early Holocene where the authors argue that there are less cedar trees due to heat stress. There are four phases where there is a clear correlation with the 18O at 10.2, 8.2, 6.0, and 5.2 (I do not see a coherence at 7.3), why are these phases not interpreted as cooler summers? Cooler water temperatures may also result in heavier calcite 18O, and could provide a different interpretation that is consistent with the cedar pollen abundance record. This may also be in line with "Atlantic cooling".

Generally, this is exactly what we said. Increased *Cedrus* pollen indicate cooler summers. However, multi-centennial changes in our $\delta^{18}O$ signal are quite large (more than 2 ‰, e.g. 8.2 ka) and the effect of temperature-dependent stable isotope fractionation during the formation of carbonate in water is not large enough for explaining these large changes. The core location is quite deep, and non-marine ostracods are (with very few exceptions) all benthic. Temperatures in modern Sidi Ali approach 8°C beneath the thermocline at 10-14 m. If temperatures were even as low as 4°C (surely not colder at lake floor because of the density maximum of water at 4°C), $\delta^{18}O$ values could have been not higher than ca. 1 ‰ (see table 1) due to the temperature change. Early Holocene $\delta^{18}O$ fluctuations are often much larger in the record and water temperature alone was surely not driving these fluctuations. Even the slighter variabilities of Late Holocene $\delta^{18}O$ values cannot be inferred from temperature-dependent stable isotope fractionation because the Sidi Ali curve shows lower values during Atlantic cooling. In the fully revised version we consider these issues and refer to Zielhofer et al. 2018b for full details.

5) I do see a possible correlation with the HSG record from Bond et al. for the Early Holocene for the positive 18O peaks around 11.4, 10.2, 8.2. However, for the peaks at 9.3, 7.3, 6.7, 6.0, and 5.2 the variation in the 18O is either very small or the timing is not comparable to the Bond-events. The timing might be due to age-model uncertainties. But in its present form, I'm unable to assess whether the Bond events and the positive peaks in 18O are within error of the age model or not, because the age uncertainties are not indicated in Fig. 2. This is definitely a must.

The age model and age uncertainties were submitted as supplementary online material in the first version of the manuscript. However, we add the error bars in the newly compiled figure (see new Fig. 3e). Further, we add lowpass filters for a better visualisation of "in phase" and "anti-phase" pattern between HSG and $\delta^{18}O$. The filtered records (500yr lowpass filter) show a good match in multi-centennial variability.

6) The 25-point running correlation calculated between the 18O and the Bond record shows correlation that barely reach 0.3, is this significant? Can you draw a line that indicates the 95% confidence level? I'm aware that age model uncertainties should also be taken into account, so this can be discussed.

We checked significance levels. Attained values above 0.4 and below approx. -0.4 are significant (95% confidence level). However, we removed the 25-point correlation in the revised version of the manuscript and argue more carefully ("in phase" vs. "out of phase").

7) During the Late Holocene the authors try to link peaks at 4.6, 4.2, 3.2, 2.7 to peaks in HSG. I truly think that this is very hard to see, because the variation in 18O is very small.

We add lowpass filters for a better visualisation of in phase and anti-phase pattern between $\delta^{18}$O and HSG records (see new Figs. 3c and 3d). The filtered records might provide a better visualisation.

8) The paper shows no figure with a comparison with regional records to test their interpretation of the ostracod 18O record, for example the pollen record from MD95-2043 (Fletcher et al., 2013) should be included. Furthermore, if the authors are correct and their 18O record represents winter precipitation, then a figure with a comparison with NAO records is necessary.

We add a NAO record in the revised manuscript (Fig. 3g; Olsen et al., 2012).

We have doubts about the value of showing MD95-2043 pollen record for comparison. The aim of our paper is to highlight a shift in phasing between $\delta^{18}$O and the Bond (2001) record, and to suggest an explanation for that. For example, during the early Holocene (high summer orbital insolation, residual ice sheets), the Bond Events were associated with strong latitudinal temperature gradient and intensification of the westerly flow and weak penetration of winter rains into the W Mediterranean. During the Late Holocene (weak summer insolation, no ice sheets, modern ocean configuration), the Bond events may be more associated with ocean current changes around the dynamics of the ocean gyres and a similar-to-present linking of cold subpolar Atlantic & NAO-like negative pattern leading to increased rainfall. The MD95-2043 shows some similarities for the early Holocene but shows a slow changing millennial behaviour for the Late Holocene that does not really help support or refute the ideas about centennial variability at Sidi Ali.

**Anonymous Referee #2** gave as important suggestions about our proxy data set. We considered his comments in the revised version of our manuscript.

Introduction and methodology: This study leads off with a thoughtful introduction reviewing and analyzing the North Atlantic (NA) rafted debris record (Bond events) and makes a strong case for Mediterranean studies showing probable linkages of hydroclimate and the Bond event record. Studies identified and compared in this work are well summarized, represent a substantial range of Mediterranean sites, and their records compared to highlight regional variability of humidity and dryness, and initially, the authors emphasize caution in attributing these patterns (in response to Atlantic cooling events) to 'forcing mechanisms, or chronological correlations'.

We thank the ref#2 for the general positive comment.

The introduction is bolstered by three well designed figures that present a broad to fine scale descriptions of the study area and place in context the North Atlantic Basin, regional climate patterns, and the coring site depicting the local landscape and vegetation.

We thank the ref#2 for the general positive comment.

The methodology was one of the strengths of this study, with the with addition of 82 new samples to a previously published 18O ostracod record for Lake Sidi Ali in the Middle Atlas range of Morocco. The new samples bring a total number of data points to 182 for 12.97 m record spanning the Early to

Late Holocene (12K cal ybp), and almost doubling the 14C chronological sample resolution of the previous record from "~130 years to 71.4 years". This robust record is reinforced by 210Pb and 137Cs dating in the historic. The 18O data were further compared with pollen (Cedrus sp.), micro-charcoal, solar activity, solar insolation, as well as a running 25-point correlation between the Bond event IRD record and the Sidi Ali 18O record. Clear figures, stacked with a color overprint of Early to Late Holocene hydroclimate changes and Bond event intervals, strongly reinforce the authors thinking.

We thank the ref#2 for the general positive comment. However, we considered comments of the ref#1 and integrated results of low-pass filtering in the figures. Further, we had to restructure the manuscript according to the comments of the ref#1

Hydroclimate: The authors characterize the overall pattern of the Sidi Ali Record in the Early Holocene with Atlantic cooling coupled with dry winters with higher summer temperatures producing drought stress limiting Cedrus. In addition, the early summer warm climate co-occurs with warm Atlantic winter rains, except during Bond events. The record is dramatically reversed, in this reviewer's opinion, for the Late Holocene beginning about 5K cal ybp where Atlantic cooling produces wet winters, in a hydroclimate of decreasing rainfall. Both Early and Late Holocene interpretations are supported from additional studies with TOC, diatom, and charcoal; and solar forcing, solar insolation, and chironomid data, respectively. This two-phase change in the Early and Late Holocene 18O record could be described as a marked low frequency, high amplitude signal that sharply decreases in amplitude after 5K cal ybp and the into the Mid-Late Holocene, and arguably begins to increase in amplitude and frequency from ~2K cal ybp into to the modern.

We thank the ref#2 for the general positive comment.

In addressing the 18O record, this reviewer suggests more description and/or insights from the authors would be helpful to interpret and the patterns of the signal with regards to amplitude and frequency, which this reader found dramatic. Possibly presenting these data with some type of signal-to-noise ratio analyses could be helpful.

Many thanks for this comment. Especially we emphasised in the discussion chapter the hydroclimatic change at ~5 ka but also the different frequencies of the d18O and Cedar records.

Additionally, this same approach could be beneficial comparing and evaluating both the pollen and charcoal data. Clearly, there is a wide range of pollen responses between the 10.2 and 7.2 Bond events, using the Early Holocene as an example. And while both responses are positive, they are clearly different in their absolute values, and appear dissimilar. I would find some characterization and analysis of this variability helpful. A similar argument can be made for the Late Holocene segment in the record, again, especially for the charcoal and pollen records. The Late Holocene charcoal signal depicts an increase in peak values of charcoal, and a variable higher frequency pattern of peaks. Some method for identifying fire events, either a threshold of signal to noise ratio,

or a confidence interval set from smoothed baseline could potentially sharpen the fire event interpretation.

Many thanks. Actually, we did not follow this suggestion in the overall revision of the manuscript because we had to restructure the manuscript following the issues of ref#1 and put the charcoal record out of the millennial-scale interpretation. Further, in the revised version the Cedrus pollen record is discussed as a probable proxy for summer temperature. We added a 1000 yr low-pass filter for a better illustration of potential coincidences between decreased subtropical summer heat (deMenocal et al. 2000) and Cedrus increases.

Finally, trends in the pollen data, such as the slower rise of Cedrus before the Early/Late Holocene shift in δ18O record, the change in the frequency of the pollen signal, and frequent occurrence of charcoal peaks beginning about 5K cal ybp also suggest there may be additional ecological factors influencing the Cedrus pollen response. Appreciated was the acknowledgment, that settlement history may have had an additional influence in the charcoal record, as is the case with many Holocene paleo-fire reconstructions.

Many thanks. Yes, we add additional thoughts about the ecological factors influencing Cedrus response. Here, the bi-millennial frequency of the Cedrus record might be evident during the Early Holocene due to the generally higher impact of summer solar radiation at that time.

4.2 Bond event: An additional strength of this study was the scholarship involved in the introduction and summary of a wide range of regional studies placing the results of the authors in context with the broader Mediterranean. Especially helpful, was the discussion of hydroclimate variability specific to other environmental reconstructions of the 4.2 period across the Mediterranean. The findings of this study showing a cool wet event at the 4.2 Bond event, was nicely contrasted with a number regional studies mostly indicating the 4.2 as a period of dryness. Further, the thoroughness of the authors placing this study in the context of such a wide spectrum of studies throughout the entire paper, potentially could be improved by a table or matrix figure summarizing this study's results and the many citations included within, bringing readers less familiar with Holocene paleo-environments of the Mediterranean region, into the sphere of thinking of the authors.

We thank the ref#2 for the general positive comment and the helpful suggestion. Right, in the final version of the manuscript we integrated a matrix figure summarising major study's results (Fig. 4).

Conclusions: This is a well written paper, and should be published with minor revisions. The amplification of the resolution of a previously published record, and subsequent interpretation of that record, is an important and detailed contribution to the understanding of paleo-hydroclimate dynamics of Morocco. In addition, the paper makes a significant case for further investigations of Holocene paleo-hydroclimate scenarios in a broader Mediterranean context, with the comparison of this record to numerous efforts, emphasizing "coherence with Bond events across the entire

Holocene" for some areas, yet in contrast, other sites demonstrating a variable step change from a wetter Early Holocene to a Late-Holocene of aridity at 5K cal ybp.

We thank the ref#2 for the general positive comment.

Post-script comments on the author response to Anonymous Reviewer 1: Figure, table, and comments submitted in both responses to Reviewer 1, greatly strengthened this submission. Especially helpful were 2 sigma age model panel, in addressing uncertainties, and the 500-year low pass filter on the pollen record and 18O reconstruction clarifying the Cedrus response.

We thank the ref#2 for the general positive comment.

Dear editor, **you** gave as additional comments and instructions:

Thanks for your patience in awaiting reviews! I find your responses to be helpful and appropriate so please prepare the final revised manuscript with the proposed changes and revised figures as outlined in your responses. It is not clear why you included Usoskin et al in one version of the figure and that does not seem to be very useful, so I suggest leaving that out or explaining its significance.

Right, we removed Usoskin from the final version of figure 2.

Similarly, the TSI plot does not add much to the paper and unless its importance can be explained, I would suggest leaving that out also.

The TSI plot is mentioned in the discussion (page 12 line 25) and we would like to show the record in Fig. 3.

All that being said, I think this is a thoughtful and interesting paper that is appropriate for publication in CoP, and as part of the "Special Issue on the 4.2ka BP event"

Many thanks for your comments and support in editing the manuscript

Kind regards

Christoph Zielhofer and all co-authors

References

[revised manuscript text omitted]